# Endogenous activity modulates stimulus and circuit-specific neural tuning and predicts perceptual behavior

Yuanning Li [1,2,3,7 ✉], Michael J. Ward[3], R. Mark Richardson[3,4,5], Max G'Sell[6] & Avniel Singh Ghuman[1,2,3]

Perception reflects not only sensory inputs, but also the endogenous state when these inputs enter the brain. Prior studies show that endogenous neural states influence stimulus processing through non-specific, global mechanisms, such as spontaneous fluctuations of arousal. It is unclear if endogenous activity influences circuit and stimulus-specific processing and behavior as well. Here we use intracranial recordings from 30 pre-surgical epilepsy patients to show that patterns of endogenous activity are related to the strength of trial-by-trial neural tuning in different visual category-selective neural circuits. The same aspects of the endogenous activity that relate to tuning in a particular neural circuit also correlate to behavioral reaction times only for stimuli from the category that circuit is selective for. These results suggest that endogenous activity can modulate neural tuning and influence behavior in a circuit- and stimulus-specific manner, reflecting a potential mechanism by which endogenous neural states facilitate and bias perception.

[1] Center for the Neural Basis of Cognition, Carnegie Mellon University and University of Pittsburgh, Pittsburgh, PA, USA. [2] Program in Neural Computation and Machine Learning, Carnegie Mellon University and University of Pittsburgh, Pittsburgh, PA, USA. [3] Department of Neurological Surgery, University of Pittsburgh, Pittsburgh, PA, USA. [4] Department of Neurosurgery, Massachusetts General Hospital, Boston, MA, USA. [5] Harvard Medical School, Boston, MA, USA. [6] Department of Statistics, Carnegie Mellon University, Pittsburgh, PA, USA. [7]Present address: Department of Neurological Surgery, University of California, San Francisco, CA, USA. ✉email: yuanningli@gmail.com

Perception depends on not only sensory input, but also the neural and cognitive state when a stimulus is presented. Traditionally, this endogenous activity has been treated as random biological noise[1]. However, studies in both humans and animals demonstrate that rather than being a noise process, endogenous activity reflects fluctuations of neural activity that influence neural processing in a behaviorally relevant manner. Specifically, endogenous fluctuations in neural activity influence both the coarse aspects of the neural response to sensory input[2–5] and the behavioral response to that input, including sensory awareness and perceptual decisions[6–12]. Endogenous activity has rich structure, reflecting the stimulus processing properties of the local neural circuitry[13], broad scale brain network architecture[14], and may reflect statistically optimal representations of the environment[15]. Fluctuations in endogenous processes such as arousal[16–18] and alertness[19,20] can influence stimulus processing and behavior. Some theoretical accounts posit that fluctuations of endogenous activity can facilitate stimulus processing in a stimulus-specific manner; for example, some have hypothesized that endogenous activity reflects predictive processes[21,22]. These theories suggest that endogenous activity can modulate the quality of the perceptual representation[15,21], as reflected in stimulus specific activity patterns or tuning. However, most studies have primarily examined nonspecific mechanisms, such as arousal and alertness[16–20]. Thus, there is a dearth of empirical evidence testing whether endogenous processes can influence neural tuning and ultimately influence behavior in a circuit and stimulus-specific manner.

Previous studies in humans have established the relationship between features of endogenous activity and the evoked response, including oscillatory phase and power/amplitude of the event-related response[23–25] or blood oxygen-level dependent (BOLD) signal[26]. "Endogenous neural state" is often operationalized in these studies as the pre-stimulus neural activity. While these studies show that pre-stimulus activity may affect the stimulus evoked response, they do not establish whether it can modulate the neural representation for stimuli in ways that are related to perception, such as the strength of tuning for particular stimuli in certain brain regions. A common way to study population-level neural tuning is to use a multivariate discriminant model to assess the separability of the population neural activity with regard to different categories[27,28]. Specifically, discriminant models extract critical dimensions in the space of the evoked response that discriminate the preferred category from the others[28]. As such, decoding accuracy may be considered a proxy for the strength of neural tuning at the population level.

In this study, we design a specific two-stage discriminant model and use intracranial electroencephalography (iEEG) recordings to test three main concrete hypotheses sequentially: (1) pre-stimulus activity modulates the decoding accuracy in response to visual stimuli; (2) the same aspect of the pre-stimulus activity that modulates decoding accuracy also correlates with behavioral perception in a region-by-stimulus specific manner, where endogenous activity in regions selective for a particular stimulus will only correlate with behavior for that stimulus (e.g. endogenous activity in regions selective for faces will correlate with behavioral performance for face stimuli); (3) the aspect of pre-stimulus activity that modulates decoding accuracy and behavior is uncorrelated across regions selective for different visual categories. Our results support these three hypotheses and suggest that endogenous fluctuations can (1) modulate stimulus-specific visual category tuning; (2) the same aspect of the activity that modulates tuning also influences behavior; (3) that this modulation does not reflect an unspecific phenomenon, such as arousal, but rather differentially and independently influences circuits selective for different categories of visual stimuli.

Additional analyses elucidate further details about what aspects of the endogenous activity modulate stimulus-specific category tuning and behavior.

## Results

**Category-selective iEEG electrodes**. Data were acquired from 30 human neurosurgical patients with implanted iEEG electrodes while they viewed grayscale images of faces, bodies, words, hammers, houses, and scrambled non-objects and performed a 1-back, repeat detection task. Electrical potentials from the iEEG electrodes and the button press reaction time (RT) for the 1-back task were recorded for all the participants. Stimuli were balanced across categories and presented in a random order to reduce any potential cognitive or strategic processes that might favor one stimulus over another. This allowed us to probe the relationship between endogenous activity, visual category tuning, and behavior separately for different categories of stimuli, and separately for the cortical circuits selective to these categories. Analyses identified 246 iEEG electrodes selective for one of these visual categories that were then used for the primary analyses examining the effects of endogenous activity on category selectivity. These iEEG electrodes were distributed across the cortex, though primarily concentrated in the bilateral ventral temporal cortex (VTC) (Fig. 1, Table 1).

**Pre-stimulus activity modulates neural stimulus decoding**. How classification accuracy in these category selective electrodes changed when the classifier was conditioned on the pre-stimulus was then examined. Specifically, single-trial potential (stP), single-trial broadband high-frequency activity (stBHA), and phases at different frequencies were extracted from the pre-stimulus activity. These aspects of the activity were used because the stP is the closest to the raw data recorded, stBHA has been shown to emphasize aspects of the activity most closely related to the underlying neuronal population firing rates[29,30], and prior studies have emphasized that pre-stimulus phase is related to stimulus response[2–5]. A two-stage model was used to modulate classification boundary based on the pre-stimulus activity (described below), and any resulting improvement in accuracy was assessed. Because the pre-stimulus activity contains no information about the upcoming stimulus category (see Supplementary Note 1), classification accuracy can be improved using this model only if the pre-stimulus activity contains information about the conditional distribution of the post-stimulus response on a particular trial (e.g. larger/lower variance, gain, etc.).

The algorithm is designed to use this pre-stimulus information, if it is present, to adjust the classification boundary, i.e. trial-by-trial tuning, on each trial to optimize classification (Fig. 1c; see "Methods" for details). Comparing classification accuracy with and without this adjustment tests the first hypothesis that endogenous activity modulates neural tuning. In addition, this adjustment provides a trial-by-trial measure of how much influence pre-stimulus activity has on neural decoding accuracy, which we term the "modulation index" (MI) (Fig. 1c). As an example, one way in which the pre-stimulus information could influence classification accuracy is to modulate the "gain" of the neural population for a particular trial. In this example, the MI would reflect the amount of gain modulation on a particular trial. Note that the algorithm is sensitive to other types of modulations, not just gain. Next the correlation between trial-by-trial MI and behavioral reaction time on a simple perceptual task was examined. This MI-reaction time correlation tests the second hypothesis that the same aspect of the pre-stimulus activity that modulates decoding accuracy also correlates with behavior. Furthermore, the correlation of the MI between pairs of

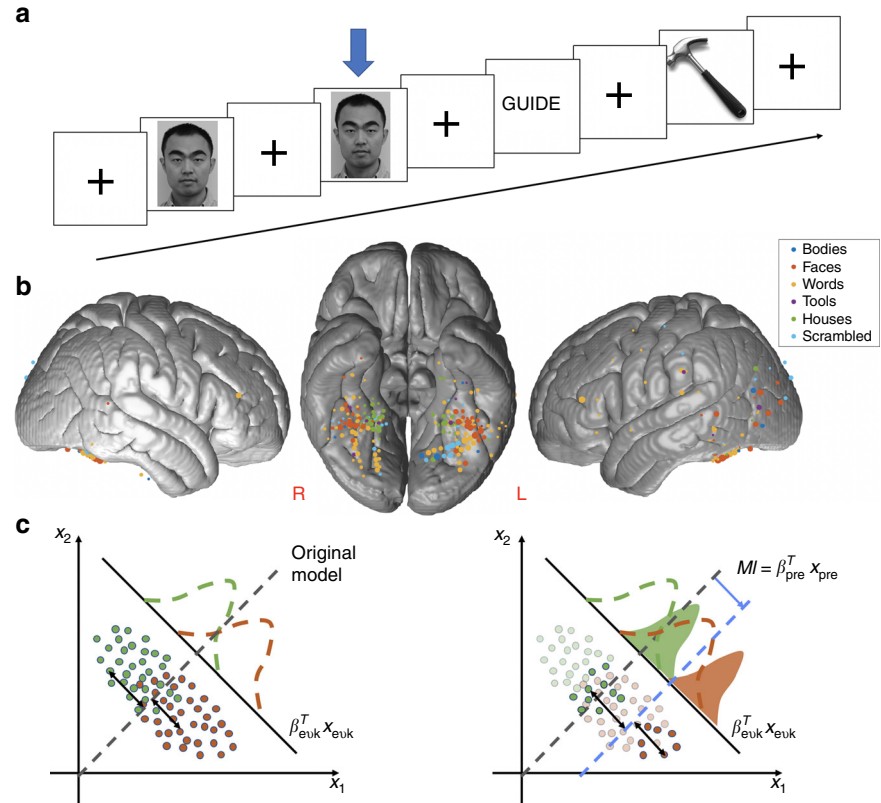

**Fig. 1 Behavioral task, the category-selective electrodes, and the algorithm framework. a** Experimental paradigm in which the subject is shown a series of images and performs a 1-back exact image repeat detection task. 180 images from 6 categories (faces, bodies, words, tools, houses, scrambled non-objects) were used. Each image was presented for 900 ms, with 900 ms inter-stimulus interval. **b** The lateral and ventral views of the locations of the 246 category-selective electrodes mapped onto a common brain surface. Color coded according to the category-selectivity of each electrode. Size of the marker indicates the relative strength of category-selectivity. The category-selectivity was determined based on (1) significant sensitivity index ($d'$) for certain category using a 6-way classifier and (2) larger event-related potential (mean stP) or broadband high-frequency activity (mean stBHA) over other categories. (**c**) Illustration of the statistical model that uses pre-stimulus to modulate trial-by-trial classification boundary. $x_1$, $x_2$ are post-stimulus features, solid black line indicates the critical discriminant dimension extracted from post-stimulus distributions that maximally separate the two categories, dashed lines indicate the classification boundary. In this algorithm, a classifier is first trained on just the post-stimulus activity (left panel, equivalent to no pre-stimulus modulation: green and orange dots are samples from two categories; dashed curves are the density plots of the two categories; dashed gray line is the post-stimulus classification boundary). The model then learns the relationship between the optimal classification boundary and the pre-stimulus activity pattern. The right panel shows an example case where the opaque dots and the solid density plots are the conditional distribution given a particular pre-stimulus activity. In this case, the optimal decision boundary (dashed blue line) is shifted to the right from the original boundary (dashed gray line). The model is sensitive to other types of shifts that may be associated with different pre-stimulus activity patters as well, such as a relationship between the pre-stimulus activity and the variance of the neural response. The size of this shift is the modulation index (MI; blue arrow).

**Table 1 Comparisons of the classification results for each category.**

| Category | Bodies | Faces | Words | Tools | Houses | Scrambled non-objects |
|---|---|---|---|---|---|---|
| # of electrodes | 9 | 56 | 92 | 16 | 37 | 36 |
| $d'$ (evoked only) | 1.1822 | 1.3957 | 0.9252 | 0.7289 | 1.0585 | 0.8219 |
| $d'$ (evoked + endogenous) | 1.3093 | 1.5072 | 1.0628 | 0.8334 | 1.2046 | 1.0105 |
| t-stat (paired t-test) | 2.7173 | 4.9466 | 7.6066 | 3.6431 | 5.1421 | 5.2407 |
| $p$-value (two-sided) | 0.0264 | $<10^{-5}$ | $<10^{-5}$ | 0.0024 | $<10^{-5}$ | $<10^{-5}$ |

electrodes that record from areas selective for the same versus different categories of visual stimuli was examined. This inter-electrode correlation tests the third hypothesis that the aspect of the pre-stimulus activity that modulates decoding accuracy is stimulus-specific and thus uncorrelated across circuits selective for different stimuli.

The results indicated that conditioning the model on pre-stimulus activity improved the classification accuracy for all visual categories, compared to the classification accuracy using only post-stimulus activity (Fig. 2a, Table 1). Mean sensitivity index $d' = 1.06$ without conditioning on pre-stimulus activity versus mean $d' = 1.19$ after conditioning on pre-stimulus activity ($t(245) = 12.39$; $p < 1 \times 10^{-10}$, paired t-test, two-sided). One potential confound is the pre-stimulus activity could reflect cognition related to the previous trial and particularly repetitions of the same condition. For example, if subjects were presented two face trials in a row, the

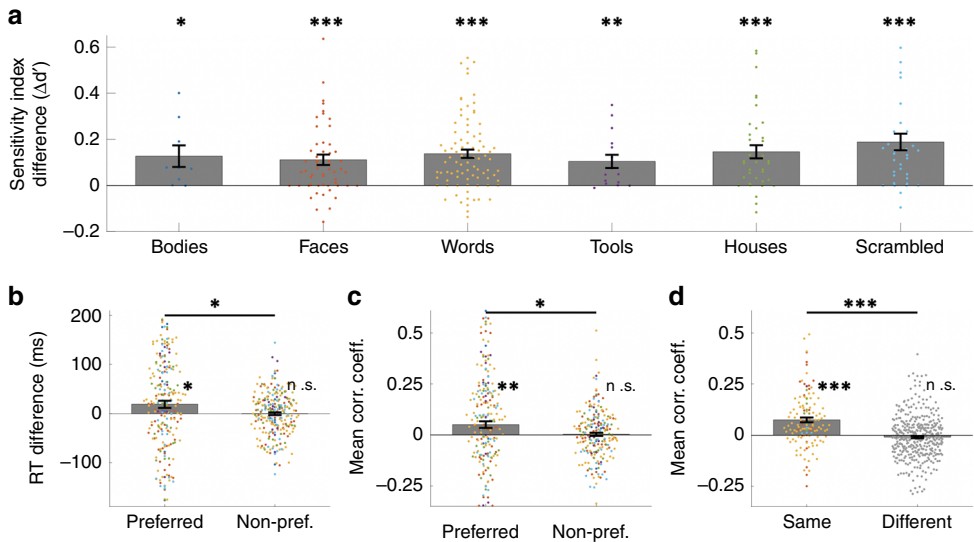

**Fig. 2 Pre-stimulus activity modulates post-stimulus tuning and predicts behavior. a** Bar plot of the averaged difference in category classification accuracy (mean sensitivity index difference $\Delta d'$) for each category. $\Delta d'$ is the accuracy conditioning on pre-stimulus activity (post-stimulus + pre-stimulus) minus the accuracy using only post-stimulus activity (post-stimulus only). Each colored dot represents the results from a single electrode ($n = 9$, 56, 92, 16, 37, 36, respectively, for each plot). See Table 1 for detailed paired comparison results for each category. **b** The difference in reaction time in between high MI trials (top quarter) and low MI trials (bottom quarter) in the preferred condition of the electrode (left) and in the non-preferred condition of the electrode (right). Each colored dot represents the averaged RT difference from one single electrode, 246 samples in total for each bar plot. **c** The correlation coefficients between modulation index (MI) and trial-by-trial reaction times at each electrode for the preferred condition of the electrode (left) and for the non-preferred conditions of the electrode (right). Each colored dot represents the correlation coefficient from one single electrode, 246 samples in total for each bar plot. **d** The correlation coefficients (Spearman's rho) for cross-electrode correlation in MI between a pair of electrodes with the same category selectivity (left) versus a pair of electrodes with different category selectivity (right). Each colored dot represents the correlation between a pair of electrodes. Data are presented as mean values ± s.e.m.; * $p < 0.05$, ** $p < 0.01$, *** $p < 0.001$, **n.s.** $p > 0.1$, two-sided paired $t$-test for (**a**) and (**c**), permutation test for (**b**) and (**d**). The dots in (**a–d**) are colored according to the corresponding selective category of the electrode, using the same color scheme as in Fig. 1b. (Source data are provided as a Source Data file.).

pre-stimulus activity for the second trial could reflect lingering activity from the first trial. This potential confound was addressed by demonstrating that classification accuracy improves with inclusion of the pre-stimulus activity, even after accounting for trial order effects, particularly repetitions of the same condition (Supplementary Table 1). These results show that critical features of pre-stimulus activity relate to the strength of neural tuning and that modifying the discriminant model based on this relationship improves classification accuracy. Therefore, these results support the first hypothesis that endogenous activity modulates the degree of category tuning in response to visual stimuli.

**Pre-stimulus activity predicts perceptual behavior.** The neural decoding accuracy is believed to reflect the quality of the neural representation[31], or population tuning, which in turn influences the quality of perception[32–34]. To make a connection between the aspect of pre-stimulus activity that modulates neural decoding accuracy and perceptual behavior, the degree to which the algorithm adjusted the classification boundary on a trial-by-trial basis was determined (the aforementioned "modulation index"; MI) and compared to behavioral reaction times. Reaction times were 18.7 ms faster on average for the bottom quarter of trials than the top quarter of trials indexed by MI for the "preferred" condition (e.g. face trials for electrodes recording from face selective regions, word trials for electrodes recording from word selective regions, etc.; $RT_{bottom} = 663.2$ ms, $RT_{top} = 681.9$ ms, $p = 0.014$, permutation test), but was not significantly different for the non-preferred condition (e.g. non-face trials for electrodes recording from face selective regions, etc.; $RT_{bottom} = 669.9$ ms, $RT_{top} = 669.6$ ms, $p > 0.1$, permutation test; Fig. 2b). Furthermore, the MI

was significantly correlated to reaction times on a trial-by-trial basis for the preferred condition (Fig. 2c; mean Spearman's rho = 0.051, $t(245) = 2.78$, $p = 0.0058$, two-sided), but not the non-preferred condition (Fig. 2c; mean Spearman's rho = 0.0031, $t(245) = 0.376$, $p = 0.71$, two-sided). The mean correlation coefficient of preferred condition was significantly larger than the mean correlation coefficient of non-preferred condition ($t(245) = 2.35$, $p = 0.02$, paired two-sided). Note that for the preferred conditions in Fig. 2b, c, only 1/6 of the repeated trials were included in the analysis (1 out of 6 categories and 20% of trials were repeat, so on average 20 trials in each block per subject per electrode). As a result, larger variance for preferred conditions was seen in Fig. 2b, c, compared to the rest of Fig. 2. These results show that the same aspect of the pre-stimulus activity that influences neural decoding accuracy in a region also correlates with the trial-by-trial response time on a perceptual task in a region-by-stimulus specific manner, which supports the second hypothesis.

One question is whether the correlation between MI and behavior reflects a general relationship between post-stimulus variability and behavior or whether the aspect of the post-stimulus discriminant activity modulated by the pre-stimulus is particularly correlated to behavior. No significant correlation was found between the loadings of the post-stimulus features in the classifier and the RT for trials in the preferred category of the electrodes, significantly lower than the correlation between the MI and the RT (mean Spearman' rho $= -0.019$ for post-stimulus features-MI correlation; comparing the pre-stimulus MI-RT correlation to the post-stimulus-RT correlation $t(245) = 2.76$, $p = 0.006$, paired two-sided). This suggests that aspect of the post-stimulus discriminant activity that is modulated by the

pre-stimulus is particularly correlated to behavior and not a result of a generic relationship between stimulus discriminant activity and behavior.

Notably, while the majority of the category-selective electrodes were located in VTC similar effects are seen in the non-VTC recordings as well. Specifically, when considering only non-temporal recordings (15 electrodes in frontal and parietal areas), the mean sensitivity index $d' = 1.00$ and 0.83 with and without conditioning on pre-stimulus activity respectively ($t(14) = 3.76$, $p = 0.002$, two-sided). Furthermore, the reaction times were 55.9 ms faster on average for the bottom quarter of trials than the top quarter of trials indexed by MI for the preferred condition (did not reach $p < 0.05$, but the effect is in the same direction as in VTC). The main results also hold if, conversely, only VTC

electrodes are considered (see Supplementary Table 2). Taken together, these results suggest that the effects of pre-stimulus activity on classification accuracy and behavior reported here are a general property of the cortex, both in VTC and outside of these regions.

**The pre-stimulus modulation is circuit-specific.** If fluctuations of endogenous activity can influence neural coding and behavior in a stimulus-specific manner, then these fluctuations should be uncorrelated across regions selective for different visual stimulus categories. In particular, endogenous fluctuations could be a reflection of changes in global cognitive state, such as arousal, or general task effects, such as changes in alertness. In these cases, the MI would correlate across the brain regions involved in the

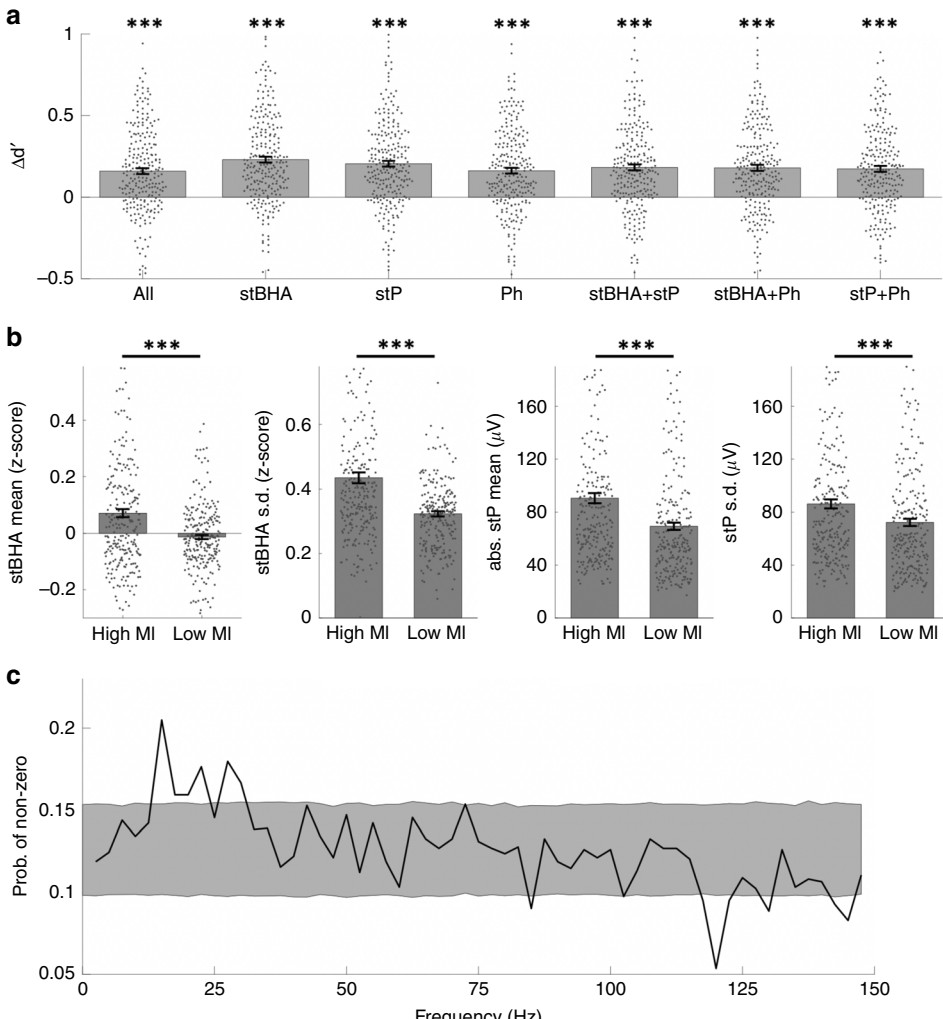

**Fig. 3 Different aspects of pre-stimulus features contributed to the modulation model. a** From left to right, bar plot for the averaged increasement classification $d'$ from post-stimulus only classification across all electrodes for: (**all**) including all pre-stimulus features, (**stBHA**) including the pre-stimulus stBHA features only, (**stP**) including the pre-stimulus stP features only, (**Ph**) including the pre-stimulus phase features only, (**stBHA+stP**) including the pre-stimulus stBHA and stP features, (**stBHA+Ph**) including pre-stimulus stBHA and phase features, (**stP+Ph**) including the pre-stimulus stP and phase features. Each sample dot in the plot represents the averaged value from one single electrode, $n = 246$ samples in total for each plot; data are presented as mean values ± s.e.m.; *** $p < 0.001$, two-sided paired $t$-test, Bonferroni correction for multiple repeated tests. (From left to right, $p = 6 \times 10^{-17}$, $3 \times 10^{-29}$, $4 \times 10^{-25}$, $7 \times 10^{-18}$, $2 \times 10^{-20}$, $4 \times 10^{-19}$, $8 \times 10^{-19}$, respectively). **b** From left to right, bar plot for distributions of: (1) the averaged z-scored stBHA power, (2) the standard deviation of z-scored stBHA power, (3) the averaged absolute value of stP, (4) the standard deviation of stP, within [−500−ms,−100 ms] pre-stimulus time window for low MI and high MI trials in each electrode. Each sample dot in the plot represents the averaged value from one single electrode, $n = 246$ samples in total for each plot; data are presented as mean values ± s.e.m., *** $p < 0.001$, permutation test. (From left to right, $p = 7 \times 10^{-6}$, $2 \times 10^{-8}$, $3 \times 10^{-23}$, $4 \times 10^{-16}$, respectively). **c** The averaged empirical probability of having non-zero weights in the sparse GLM model for different pre-stimulus phase features of different frequency. Shaded area indicates 95% confidence interval under random feature selection. (Source data are provided as a Source Data file.).

task, regardless of category-selectivity of a particular region. However, cross-electrode correlation in MI was weakly, though statistically significantly, correlated only between electrodes that share the same category-selectivity (mean Spearman's rho = 0.076, $p < 0.001$, permutation test) and not significantly correlated for electrodes of different category-selectivity (mean Spearman's rho = −0.0092, $p > 0.1$, permutation test; Fig. 2d). Significantly larger correlation was seen between electrodes of the same category-selectivity than electrodes of different category-selectivity ($t(520) = 7.54$, $p < 1 \times 10^{-10}$, two-sided; Fig. 2d). As a result, the pre-stimulus modulation is partially a reflection of weakly correlated fluctuations within category-specific networks, but it does not seem to reflect non-specific processes, such as arousal or alertness, because correlations are not seen across all category selective electrodes. The network specific, but not global, correlation supports our third hypothesis.

**Activity features that contribute to pre-stimulus modulation.** The results above support our three major hypotheses and shows that pre-stimulus activity can influence neural tuning and behavior in a stimulus-specific manner. A number of questions regarding the nature of the pre-stimulus activity that influences decoding accuracy and perception remain. To evaluate the contribution of different aspects of the pre-stimulus features, the same model was applied using different subsets of the pre-stimulus features. This analysis showed that the pre-stimulus stP, which is dominated by the low frequency component, the pre-stimulus stBHA, which reflects the power of high frequency broadband activity, and the pre-stimulus oscillatory phase all contributed to the modulation of category decoding accuracy, with the stBHA showing the highest increase in accuracy (Fig. 3a). The trials were then ranked by their MI and the mean and standard deviation in their pre-stimulus stP and stBHA were compared. The bottom quarter of trials had significant lower mean and variance for both stBHA and absolute stP during the pre-stimulus period, compared to the top quarter of trials (Fig. 3b). Given that lower MI trials correspond to shorter RTs, the decreased pre-stimulus mean and variance may be an indication of lower pre-stimulus noise[35] or fluctuations of stimulus-specific attention[36], which leads to shorter RTs. A further analysis into the distribution of non-zero weights in the two-stage GLM suggests that the alpha/beta phases, from 10 to 25 Hz with a peak at 15 Hz, showed a consistent pattern of modulation of category decoding accuracy (Fig. 3c), suggesting a role for the phase of endogenous oscillations in this frequency range when visual stimuli are presented. Recent studies have shown that deployment of endogenous attention reflects neural coherence in a similar frequency range as the one seen in the current study[37], suggesting the pre-stimulus facilitation seen here may reflect fluctuations of endogenous attention.

**The temporal scale of the modulation effect.** Previous studies have shown that infra-slow fluctuations of activity, seen in "resting-state" studies, are associated with fluctuations of behavior and perception[38–40]. If the aspect of the pre-stimulus activity that modulates neural decoding accuracy and behavior seen in the present study reflected these intra-slow fluctuations, there would be significant auto-correlation within each channel between consecutive trials for the MI. The auto-correlation of MI across consecutive trials for each electrode was computed, and 40 out of the 246 electrodes (~15%) showed significant auto-correlation across trials at $p < 0.05$ uncorrected level (Fig. 4). While this is significantly more than would be expected by chance, it is a relatively small subset of the electrodes, suggesting that there is a

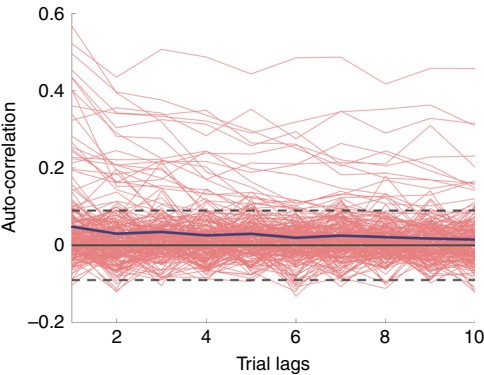

**Fig. 4 The pre-stimulus modulation effect is mostly transient.** The temporal auto-correlation for MI across consecutive trials in each category-selective channel. The blue solid line indicates the average auto-correlation across all electrodes. 40/246 electrodes showed significant auto-correlation ($p < 0.05$, uncorrected, two-sided $t$-test). The dashed lines correspond to $p = 0.05$ threshold, uncorrected. (Source data are provided as a Source Data file.).

mix of infra-slow and transient effects in the pre-stimulus activity, with transient effects being the dominant proportion.

**Discussion**

These results demonstrate that pre-stimulus activity modulates the degree of category tuning on the trial-by-trial basis in category-selective areas of the cortex. Previous studies have mainly focused on the overall correlation between the pre-stimulus activity and the evoked response in features including phase and oscillatory power of the event-related response[23–25], often not definitively localized to the regions that process the stimulus class being presented[41]. The results here demonstrate that pre-stimulus activity modulates the post-stimulus activity in the regions that are selective for the stimulus being viewed.

The results also show that the degree of influence on the neural classification accuracy from the pre-stimulus activity correlates with behavioral reaction time specifically for the category of stimulus that a particular region processes. Prior work has shown that different aspects of the pre-stimulus activity, including phase and amplitude of the event-related response/field[7,42], as well as BOLD signal[8,43], correlate to behavioral performance. However, prior studies leave unclear whether the same aspects of pre-stimulus activity that modulate category-tuning also give rise to the influence on behavior as these two different effects have mostly not been linked to one another. The results here demonstrate that the two processes can be attributed to the same aspects of pre-stimulus activity in the same local category-sensitive circuit. The results demonstrated a significant relationship between the MI and the reaction time in detecting repetitions in the category that the electrode is selective for. Furthermore, no significant correlation was found between the MI and the reaction time with respect to categories that the electrode is not selective for, suggesting that this effect is not global and non-specific, such as reflecting arousal or alertness, but restricted to specific functional neural circuits.

Taken together these results suggest a model for how endogenous states can influence neural activity to modulate the perception of specific visual stimuli. If the stimulus is presented when endogenous activity in regions selective to that type of stimulus is relatively low, as indicated by lower pre-stimulus mean and variance, and when the phases of endogenous oscillations in the alpha/beta frequency range are optimal, then neural tuning will be stronger and perceptual behavior will be facilitated. The size of

the behavioral modulation with endogenous activity (~20 ms) is on par with the magnitude of the behavioral facilitation seen with certain kinds of visual priming[44] and endogenous visual attention[45], suggesting that while the effect may be relatively small, it may play an important role in perception. The results of the present study cannot completely exclude the possibility that the behavioral correlation seen is due to endogenous activity modulating decision processes rather than perceptual processes. However, most of the electrodes examined were located in VTC regions associated with visual perception. Furthermore, the multivariate pattern of activity in VTC, which is the same aspect of the signal that pre-stimulus activity modulates in this study, has previously been linked to the subjective perceptual representation[46]. The location of the electrodes and the aspects of the neural signals examined suggest that the perceptual rather than decision processes were influenced by endogenous activity here.

Given the random stimulus presentation in the present study, facilitating one stimulus over another on a trial-by-trial basis does not provide a behavioral advantage. Therefore, it is unclear if the endogenous activity seen here reflects stochastic dynamics in brain circuits, such a fluctuations of neurotransmitter levels[47], or strategic processes, such as fluctuations in stimulus-specific attention or preference[48], that may reflect pattern detection and strategies primates adopt even when stimuli are presented randomly[49]. While in the present study a strategic process would not provide a behavioral advantage, for example visual perception in familiar environments that one commonly finds oneself in, such as one's house or office, facilitating the processing of particular stimuli may be advantageous. In these contexts, the stimulus-specificity of endogenous optimization may reflect a prediction of the next stimulus viewed based on internal models of the environment[22]. The magnitude of the effects seen here may be larger in cases where facilitating a particular stimulus over another was behaviorally useful. Active sensing in natural settings may organize the processes that underlie this optimization[50] and/or these active processes may synchronize to fluctuations in endogenous activity so that deployment of overt and covert attention occurs at temporally optimal times for information gathering[51].

One hypothesis about how endogenous fluctuations modulate neural responses and behavior is that they may reflect a priming-like pre-activation of a predicted stimulus[22], for example a prior in the Bayesian sense[52]. However, pre-activation would likely correspond to a higher pre-stimulus response in regions that process a particular stimulus type, not lower as was seen here. Without single unit recording we cannot fully exclude the alternative possibility that the reduced mean and variance of the pre-stimulus activity could be a result of desynchronization that can go along with enhanced frequency of action potentials[53,54]. However, prior studies in early visual cortex in monkeys also showed that lower pre-stimulus activity is associated with improved tuning and behavior, though in a non-specific manner associated with attentiveness[19,20,55]. The results of the present study suggest that the effects seen in early visual cortex in single units in monkeys may also occur in a stimulus- and circuit-specific manner in higher-level visual regions and in regions outside of visual cortex in humans. Lower pre-stimulus mean and variance may reflect an optimization of the dynamic range or gain[55], potentially through normalization[56] in neural circuits responsible for processing particular stimulus types to enhance information pick-up for those stimuli[57]. While reduced pre-stimulus activity and variance is not consistent with a priming-like prior, the results here do provide a potential foundation for endogenous activity to reflect predictive processing[21], though through a non-priming mechanism, such as circuit-specific optimization of processing. Note that while these results are not sufficient evidence of predictive processing, for the hypothesis

that endogenous activity is a signature of predictive coding to be correct[22], stimulus and circuit level modulation of tuning is a necessary feature of endogenous activity. The results here provide evidence of this necessary (though not fully sufficient) feature of endogenous activity needed for this activity to reflect predictive processes.

One methodological note about this work is that the two-stage statistical model described here has potential applications beyond examining the effects of pre-stimulus activity on discriminant information. Specifically, this method can be used to examine the effects of any multivariate signal on local discriminant information on a trial-by-trial basis. For example, this algorithm could be used to examine how activity in one region modulates discriminant information in another region, a form of multivariate functional connectivity[58,59]. Much like MI here, using this method for multivariate functional connectivity would yield a trail-by-trial measure of how much one region influences the representation in another region. That trial-by-trial measure of interregional influence could then be correlated to external variables, such as behavior, as was done in the present study between MI and reaction time.

Taken together, our results provide empirical support for a mechanism in which the present neural state influences the perception of sensory input in a stimulus-specific manner by modulating the tuning properties of neural circuits selective for those stimuli.

## Methods

**Subjects**. The experimental protocols were approved by the Institutional Review Board of the University of Pittsburgh. Written informed consent was obtained from all participants. 30 human subjects (11 male, 19 female) underwent surgical placement of subdural electrocorticographic electrodes or stereotactic electro-encephalography (together electrocorticography and stereotactic electro-encephalography are referred to here as iEEG) as standard of care for seizure onset zone localization. The ages of the subjects ranged from 19 to 64 years old (mean = 38.2, SD = 11.9). None of the participants showed evidence of epileptic activity on the electrodes used in this study nor any ictal events during experimental sessions.

**Stimuli**. In each session, 180 images of faces (50% male), bodies (50% male), words, hammers, houses, and phase scrambled faces were used as visual stimuli. Each of the six categories contained 30 images, and each image was presented twice. At random, 1/3 of the time an image would be repeated, which yielded 480 independent trials in each session.

**Paradigms**. In the experiment, each image was presented for 900 ms with 900 ms inter-trial interval during which a fixation cross was presented at the center of the screen (all images were scaled such that their longest dimension subtended approximately 10° of visual angle). Participants were instructed to press a button on a button box when an exact image was repeated (1-back image repeat not category repeat), and their reaction time (RT) was recorded as the period from stimulus onset until the button press in the 1-back task. Paradigms were programmed in MATLAB[TM] using Psychtoolbox and custom written code. All stimuli were presented on an LCD computer screen placed approximately 150 cm from participants' heads.

**Data preprocessing**. The electrophysiological activity was recorded using iEEG electrodes at 1000 Hz. Common reference and ground electrodes were placed subdurally at a location distant from any recording electrodes, with contacts oriented toward the dura. The 60 Hz line noise was removed using a fourth order Butterworth filter with 55–65 Hz stop-band. Single-trial field potential (stP) signal was extracted by band-passing filtering the raw data between 0.2 and 115 Hz using a fourth order Butterworth filter to remove slow and linear drift, and high frequency noise. The stP signal was sampled at 1000 Hz.

The single trial broadband high-frequency (stBHA) activity was defined as the mean z-scored PSD across 40–100 Hz on each trial. Specifically, power spectrum density (PSD) at 2–100 Hz with bin size of 2 Hz and time-step size of 10 ms was estimated for each trial using multi-taper power spectrum analysis with Hann tapers, using FieldTrip toolbox[60]. For each channel, the PSD at each frequency was z-scored with respect to the mean and variance of the baseline activity between experimental runs to correct for the power scaling over frequency at each channel. The stBHA was sampled at 100 Hz.

We define the neural activity within the [−500, −100] ms interval relative to the stimulus onset as the pre-stimulus activity, and the neural activity within the

[100, 500] ms interval relative to the stimulus onset as the post-stimulus activity. Therefore, we have 400 stP features and 40 stBHA features for both the pre-stimulus and post-stimulus activity. Specifically, there is potential signal leakage caused by the low-pass and band-pass filters, including hardware filters. Thus, we exclude the time interval around stimulus onset out of an abundance of caution to ensure there was no spillover of pre-stimulus activity into the activity used as the stimulus response (and vice versa) in a conservative manner.

The pre-stimulus phase information was also extracted from each trial. Specifically, discrete time Fourier transform was applied to the raw signal in the [−500, −100] ms time interval, which had a total length of 400 points sampled at 1000 Hz. As a result, the phase information between 0 and 1000 Hz was extracted with a step-size of 2.5 Hz. The phases from 0 to 150 Hz were used as the pre-stimulus phase features yielding 60 phase features (60 = 150/2.5).

To reduce potential artifacts in the data, raw data were inspected for ictal events, and none were found during experimental recordings. Trials with maximum amplitude 5 standard deviations above the mean across all the trials were eliminated. In addition, trials with a change of more than 25 µV between consecutive sampling points were eliminated. These criteria resulted in the elimination of less than 1% of trials.

**Electrode localization.** Coregistration of grid electrodes and electrode strips was adapted from the method of Hermes, Miller[61,62]. Electrode contacts were segmented from high resolution post-operative CT scans of patients coregistered with anatomical MRI scans before neurosurgery and electrode implantation. The Hermes method accounts for shifts in electrode location due to the deformation of the cortex by utilizing reconstructions of the cortical surface with FreeSurfer™ software and co-registering these reconstructions with a high-resolution post-operative CT scan. SEEG electrodes were localized with Brainstorm software[63] using post-operative MRI co-registered with pre-operative MRI images. Specifically, electrodes from the VTC were the ones located in inferior temporal gyrus, fusiform gyrus, parahippocampal gyrus and the sulci in between.

**Electrode selection.** Category-selective electrodes were selected based on a 6-way classifier. Specifically, we trained a multinomial logistic regression model to classify the post-stimulus neural activity with respect to the six different categories from each other. The true positive rate and false positive rate for each category were estimated using 5-fold cross-validation. The sensitivity index ($d'$) for each category was then computed as $d' = Z(\text{true positive rate}) − Z(\text{false positive rate})$, where $Z(x)$ is the inverse function of the cumulative density function of standard normal distribution. An electrode was selected as category-selective if the maximum $d'$ across all categories is greater than 0.5 ($p < 0.01$, permutation test). The selected electrode was then assigned to the category with maximum $d'$. To avoid the rare case where an electrode showed visual response to all but one category, we add additional constraint that the assigned category should have larger event-related potential (mean stP) or broadband high-frequency activity (mean stBHA) over other categories.

**Two-stage generalized linear model (GLM).** We considered the neural activity within the [−500, −100] ms pre-stimulus time interval as proxy for the endogenous activity, noted as $X_{pre} \in \mathbb{R}^{N \times T_1}$, where $N$ is the number of trials and $T_1$ is the number of features in the pre-stimulus time window; and we used neural activity from the [100, 500] ms time interval relative to stimulus onset as the post-stimulus evoked activity that encodes category information, noted as $X_{evk} \in \mathbb{R}^{N \times T_2}$, where $T_2$ is the number of features in the post-stimulus time window.

We designed a two-stage regularized GLM (logistic regression) model to evaluate the pre-stimulus modulation on category representation (see below for the pseudocode of the main algorithm). A logistic regression model predicts the category of the stimulus $y$ using a linear combination of the neural features $X$ and a logistic function[64]. Specifically, $P(y = 1|X) = 1/(1 + \exp(−X\beta))$.

In the first stage, logistic regression was directly applied to the post-stimulus activity to extract the critical discriminant dimensions ($\beta_{evk}$) for category classification. In other words, we solved for

$$\beta_{evk}^* = \underset{\beta_{evk}}{\arg\min}\ \ell(\beta_{evk}) + \lambda_1 P_\alpha(\beta_{evk}), \qquad (1)$$

where $\ell(\beta_{evk}) = −y^T X_{evk}\beta + 1^T \log(1 + \exp(X_{evk}\beta))$ is the cross-entropy loss for logistic regression[64], and $P_\alpha^{evk}(\beta_{evk}) = \frac{(1-\alpha)}{2}\|\beta_{evk}\|_2^2 + \alpha\|\beta_{evk}\|_1$ is the standard elastic-net penalty term to account for the high-dimensional settings, and $\alpha$ is the elastic-net mixing parameter that balancing between ridge (L2 term) and lasso (L1 term) regularization[65]. In our case we set $\alpha = 0.95$ to have a penalty that is primary lasso. $\lambda_1$ is the regularization hyperparameter to be chosen based on cross-validation (see below for a detailed description of cross-validation). This first stage results in a trial-by-trial neural metric, $X_{evk}\beta_{evk}$, which corresponds to the signed distance to the classification boundary and quantifies the post-stimulus category selectivity.

In the second stage, we fixed the optimal dimension $\beta_{evk}^*$ and optimized the model to modulate classification boundary along the critical discriminating directions found in the first stage, based on the pre-stimulus activity. Specifically, we solved

$$\beta_{pre}^* = \underset{\beta_{pre}}{\arg\min}\ \ell(\beta_{evk}^*, \beta_{pre}) + \lambda_2 P_\alpha(\beta_{pre}) \qquad (2)$$

where $\ell(\beta_{evk}^*, \beta_{pre}) = −y^T(X_{evk}\beta_{evk}^* + X_{pre}\beta_{pre}) + 1^T \log(1 + \exp(X_{evk}\beta_{evk}^* + X_{pre}\beta_{pre}))$, and $P_\alpha^{pre}(\beta_{pre})$ is a similar elastic-net penalty but with group structure to account for the phase features (see below for a detailed description of the penalty structure). $\lambda_2$ is the regularization hyperparameter to be chosen based on cross-validation (see below for a detailed description of cross-validation). This stage provides a neural metric $X_{pre}\beta_{pre}$ in pre-stimulus activity that quantifies the amount of influence from pre-stimulus activity on the post-stimulus category selectivity on a trial-by-trial basis. We defined $MI = X_{pre}\beta_{pre}$ as the pre-stimulus modulation index (MI).

**The (group) elastic-net penalty.** For the post-stimulus part, we only considered the stP and stBHA features, noted as $x_{evk} = [x_{evk}^P, x_{evk}^{BHA}]$, with the corresponding weights $\beta_{evk} = [\beta_{evk}^P, \beta_{evk}^{BHA}]$, and we applied regularization term $P_\alpha^{evk}(\beta_{evk}) = \frac{(1-\alpha)}{2}\|\beta_{evk}\|_2^2 + \alpha\|\beta_{evk}\|_1$ in (2). For the pre-stimulus part, we used stP, stBHA and phase features, noted as $x_{pre} = [x_{pre}^P, x_{pre}^{BHA}, x_{pre}^{phase}]$, and the corresponding weights $\beta_{pre} = [\beta_{pre}^P, \beta_{pre}^{BHA}, \beta_{pre}^{phase}]$. Assume that we have phase $[\theta_1, ..., \theta_K]$, where $\theta \in (−\pi, \pi]$, corresponding to frequencies of interest $[f_1, ... f_K]$. To transfer the circular phase value onto the real axis in order to facilitate the $\ell_1$-norm penalty, we consider feature vector $x_{pre}^{phase} = [\sin\theta_1, \cos\theta_1, ... , \sin\theta_K, \cos\theta_K]$, where $\sin\theta, \cos\theta \in [−1, 1]$, and group lasso penalty term $\mathcal{G}(\beta_{pre}^{phase}) = \sqrt{2} \sum_{i=1}^K \sqrt{(\beta_{pre,(i,1)}^{phase})^2 + (\beta_{pre,(i,2)}^{phase})^2}$, where $[\beta_{pre,(i,1)}^{phase}, \beta_{pre,(i,2)}^{phase}]$ are the pair of weights corresponding to phase feature pair $[\sin\theta_i, \cos\theta_i]$. This group structure would ensure that the penalty is invariant to the overall direction of the phase, which a typical $\ell_1$-norm penalty would not do. As a result, the group elastic-net penalty for the pre-stimulus weights can be written as

$$P_\alpha^{pre}(\beta_{pre}) = \frac{(1-\alpha)}{2}\|\beta_{pre}\|_2^2 + \alpha\|\beta_{pre}^P\|_1 + \alpha\|\beta_{pre}^{BHA}\|_1 + \alpha\mathcal{G}(\beta_{pre}^{phase}).$$

**Nested cross-validation (CV) procedure.** When fitting the two-stage GLM, we used a nested cross-validation procedure with two levels of CV. At the first level of CV (CV-1), we randomly split the data evenly into 5 folds. For each CV iteration, one fold of the data was used as the testing set while the other four folds were used as the training set. The second level of CV (CV-2) was applied to the training set to fit hyperparameters in the model. We further split the training set into 10 folds and select hyperparameters based on minimizing the overall deviance, i.e. the loss function $\ell$ in equations (1) and (2), using block coordinate descent algorithm[66] in CV-2. Specifically, this procedure was repeated two times to select $\lambda_1$ and $\lambda_2$ sequentially. After selecting the optimal hyperparameters, the entire training set was used to fit the two-stage GLM with the optimal hyperparameters. Then the fitted model was applied to the testing set to estimate the classification accuracy and single-trial MI. By iterating through all 5 folds at CV-1, we can get estimated single-trial MI for every single trial when they were included in the testing set. This procedure was applied to all cases of analysis where the two-stage GLM was used in order to get an unbiased estimation of the single-trial MI and the overall classification accuracy. The flow of the entire algorithm is shown below.

**Training and evaluating the two-stage GLM.** Here we describe in pseudocode the overall flow of training and evaluating the two-stage GLM using a nested cross-validation.
Data:
 data matrices $X_{pre} \in \mathbb{R}^{N \times T_1}$, $X_{evk} \in \mathbb{R}^{N \times T_2}$ for pre-stimulus and post-stimulus data, and the corresponding data category label $y \in \{0,1\}^N$,
 where $T_1 = [t_{pre}^P, t_{pre}^{BHA}, t_{pre}^{phase}]$, $T_2 = t_{evk}^P + t_{evk}^{BHA}$,
 and $X_{pre} = [X_{pre}^P, X_{pre}^{BHA}, X_{pre}^{phase}]$, $X_{evk} = [X_{evk}^P, X_{evk}^{BHA}]$
Hyper-parameters:
 the elastic-net mixing parameter $\alpha$ (set as 0.95), regularization parameters $\lambda_1$ and $\lambda_2$
Output:
 weight vectors $\beta_{pre}^* = [\beta_{pre}^P, \beta_{pre}^{BHA}, \beta_{pre}^{phase}]$, $\beta_{evk}^* = [\beta_{evk}^P, \beta_{evk}^{BHA}]$,
 and the corresponding categorical classification accuracy $d'_{pre}, d'_{evk}$
CV-1: Split the data into 5 folds; CV-2: Split the training set in CV1 into 10 folds.
**for** each CV-1 split $\{X_{train}^{CV1}, X_{test}^{CV1}\}$ **do**:
 *Step-1*: fit the elastic-net problem for post-stimulus features:
 **for** the $i$th CV-2 split of $X_{train}^{CV1}$ : $\{X_{evk,train}^{(i)}, X_{evk,test}^{(i)}\}$ **do**:
 **for** $\lambda_1 \leftarrow \lambda_{max}$ **to** $\epsilon\lambda_{max}$ (decrement $\lambda_1$) **do**:

- solve elastic-net problem (1) using coordinate descent; (see Supplementary Methods for details)
- estimate the deviance of the solution on $X_{evk,test}^{(i)}$;

- find the optimal $\lambda_1$ using CV-2 average, and solve (1) for $\beta_{evk}^*$ using the entire $X_{train}^{CV1}$;
- evaluate $d'_{evk}$ on $X_{test}^{CV1}$.

*Step-2*: fix $\beta_{evk}^*$ and fit group elastic-net problem for pre-stimulus features:

**for** the $i$th CV-2 split of $X_{train}^{CV1}$ : $\left\{ X_{pre,train}^{(i)}, X_{pre,test}^{(i)} \right\}$ **do**:

**for** $\lambda_2 \leftarrow \lambda_{max}$ to $\varepsilon\lambda_{max}$ (decrement $\lambda_2$) **do**:

- solve group elastic-net problem (2) using block coordinate descent; (see Supplementary Methods for details)
- estimate the deviance of the solution on $X_{pre,test}^{(i)}$;

- find the optimal $\lambda_2$ using CV-2 average, and solve (2) for $\beta_{pre}^*$ using the entire $X_{train}^{CV1}$;
- evaluate $d'_{pre}$ on $X_{test}^{CV1}$;
- The modulation index (MI) is estimated as $MI = X_{pre,test}^{CV1}\beta_{pre}^*$.

- Estimate the overall mean $d'_{pre}$ and $d'_{evk}$ based on 5-fold CV-1.

**Cross-electrode correlation in pre-stimulus MI**. To evaluate the spatial properties of the pre-stimulus modulation effect, we computed the correlation of the single trial pre-stimulus MI between category-selective electrodes in each subject. For the $i$th category-selective electrode, we got $MI_i = X_{pre,i}\beta_{pre,i}$ from the GLM. The cross-electrode correlation between two category-selective electrodes $i$ and $j$ was estimated by computing the correlation coefficient between $MI_i$ and $MI_j$ across all trials. To avoid confounding effect from local spatial correlation between two nearby electrodes, we only considered a pair of electrodes that were > 2 cm apart from each other. For each subject, the mean cross-electrode correlation was estimated by averaging the pairwise correlation coefficients across all such pairs of category-selective electrodes.

Permutation testing was used to test for significance of the cross-electrode MI correlations (Fig. 2d). Specifically, for each permutation, we randomly shuffled the category condition of all the trials and repeat the above analysis to compute the mean cross-electrode correlation coefficients for electrodes with the same/different category selectivity. This process was repeated for 1000 times to get the histogram of the null distribution of the averaged correlation coefficient.

**Autocorrelation in MI**. To evaluate the temporal properties of the pre-stimulus modulation effect, we computed the autocorrelation of the single trial MI between consecutive trials with lags ranging from 1 to 20 in each category-selective electrode. Specifically, for any given electrode, the autocorrelation with lag $k$ is $r_k = \frac{\sum_{t=1}^{T-k}\left(MI^{(t)} - \overline{MI}\right)\left(MI^{(t+k)} - \overline{MI}\right)}{\sum_{t=1}^{T}\left(MI^{(t)} - \overline{MI}\right)\left(MI^{(t)} - \overline{MI}\right)}$. To evaluate the temporal property, we tested for the significance of the first-order autocorrelation, since it is essential for any temporal dependencies caused by slow-fluctuation in the signal. Specifically, the upper bound of the 95% confidence interval was approximately estimated as $2/\sqrt{T}$ where $T$ is the total number of trials.

**Permutation test for differences based on high vs low MI**. Permutation test was used to test for significance of the differences in RT, pre-stimulus stP, and pre-stimulus stHBA based on MI in this study (Fig. 2d, Fig. 3b). In order to construct a surrogate distribution of the MI, in the $i$th permutation we generated random projection weight vector $\beta^{(i)} = \left[\beta_1^{(i)}, \ldots, \beta_{T_1}^{(i)}\right] \in \mathbb{R}^{T_1}$, such that $\left\|\beta^{(i)}\right\|_0 = \left\|\beta_{pre}\right\|_0$. Specifically, let $n = \left\|\beta_{pre}\right\|_0$, we randomly drew $\{p_1, \ldots, p_n\} \subset \{1, \ldots, T_1\}$, and then $\beta_p^{(i)} \sim N(0,1)$ if $p \in \{p_1, \ldots, p_n\}$, $\beta_p^{(i)} = 0$ otherwise. Then we computed $MI = X_{pre}\beta^{(i)}$ and sorted the trials according to this permuted MI in order to compute the differences in RT, pre-stimulus stP, and pre-stimulus stHBA between trials in the top quarter and trials in the bottom quarter. We repeated this process 1000 times for each electrode, and the histograms of those differences were used as the null distributions based on permuted MI.

**Correlation between post-stimulus discriminant activity and behavior**. We used the loadings in Step 1 of the two-stage model, i.e. $X_{post}\beta_{post}$, as the neural metric for post-stimulus discriminant activity. The linear correlation between $X_{post}\beta_{post}$ and $RT$ for each repeated trial in the preferred condition was estimated in each electrode.

**Reporting summary**. Further information on research design is available in the Nature Research Reporting Summary linked to this article.

## Data availability

The dataset generated during the current study will be made available from the authors upon reasonable request. A reporting summary for this article is available as a Supplementary Information file. The source data underlying Figs. 1–4 are provided as a Source Data file.

## Code availability

A demo code of the main algorithm with sample data can be found at https://github.com/yuanningli/two-stage-GLM. The completely developed code that operates on the full dataset will be made available from the authors upon reasonable request.

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

## Acknowledgements
The authors would like to thank the patients and staff in the epilepsy monitoring unit at the University of Pittsburgh Medical Center for their participation in this research study. We would also like to thank Shawn Walls, and Ashley C. Whiteman for their help with data collection. We thank Charles Schroeder and Matthew Smith for critically reading the manuscript and for helpful suggestions. The authors gratefully acknowledge the support of the National Institute of Mental Health under R01MH107797 (to A.S.G.), R01MH064537 (to M.G.), and RF1MH114223 (to A.S.G.), National Eye Institute under R21EY030297 (to A.S.G.), and National Science Foundation under 1734907 (to A.S.G. and M.G.).

## Author contributions
Conceptualization, Y.L. and A.S.G.; Methodology, Y.L., M.G., and A.S.G.; Investigation, Y.L., M.J.W., R.M.R., and A.S.G.; Formal Analysis, Y.L., M.G., and A.S.G.; Writing—Original Draft, Y.L. and A.S.G.; Writing—Review & Editing, Y.L., M.G., R.M.R., and A.S.G.; Resources, R.M.R. and A.S.G.; Funding Acquisition, A.S.G.

## Competing interests
The authors declare no competing interests.
