## [Peer Review File · Nature Communications]

Reviewers' Comments:

Reviewer #1:

Remarks to the Author:

In their manuscript Li et al. address the interesting question to what extent "endogenous" (rather prestimulus) activity shape visual category specific neural responses and whether these -traditionally usually neglected- neural pattern are also behaviorally relevant. To this end they use iEEG data recorded from 30 epileptic patients and apply a multivariate regression approach to first identify classification based on post-stimulus data and then adding -in a second step- prestimulus data in order to test whether classification accuracy (d-prime measure) is improved. From the latter analysis they are able to quantify the extent to which prestimulus information changes model weights and use this measure (\sim Modulation Index) to follow up with further analysis on a single-trial level. Among other aspects they are able to show that the MI measures correlate between electrodes with similar category sensitivity, whereas they are virtually uncorrelated between electrodes that are sensitive to different categories. Also the authors claim that lower prestimulus activity (\sim lower LFP and gamma band power) go along facilitatory / biasing effects.

Overall there is a lot that I like about this study, even though I admit to struggle with the details of the methodological approach and will freely admit that I probably cannot judge whether all is done in a satisfactory manner. However, also given the chosen "general interest" format of the journal, I think the authors should make a stronger effort in making their approach more accessible in general terms at strategically well-chosen points (outside of the methods section). Furthermore, I think that central claims -at this stage- are not well backed up: especially the aspect that the (prestimulus) results support an improved neural tuning at a perceptual, rather than at a decision level. Thus for the moment the manuscript requires significant improvements before publication can be considered.

Issues:

* The most crucial issue in my opinion is that the authors intend to communicate that their prestimulus patterns bias "neural tuning" supporting perceptual processing. This is not clearly shown and the effects (neurally, but also the RT effects) could equally well be biasing decision making processes. In fact many motor and prefrontal electrodes appear as category sensitive (Figure 1). While I do not exclude the possibility that they may be relevant for category perception, the fact that the task involved (speeded) motor responses does not help in disambiguating these different contributing influences.

* Regarding the interpretation that "low activity" biases neural tuning and behavior in category specific manners, I am not sure to what extent this claim can be made in a strong manner. This pattern could in principle be driven by regionally specific power desynchronizations (which can be regionally extremely specific; <https://www.ncbi.nlm.nih.gov/pubmed/31291043>), that can go along with enhanced frequency of action potentials (<https://www.ncbi.nlm.nih.gov/pubmed/22084106>). Thus, the results could in principle support the quite opposite conclusion.

* As stated above, I will not pretend to understand all details of the methodological approach. However with solid M/EEG background I would consider myself a representative reader that is interested in the general topic. I would really welcome a better effort to take the reader by the hand to understand the critical elements. This would include some helpful "fool-proof" comments here and there, but also I think a Figure (e.g. added to Figure 1) visualizing some aspects (e.g. how the MI is derived, which is critical for several conclusions). The methods descriptions makes me assume that regression models were computed for every trial and electrode separately, is this correct (also asking because this is not a mainstream approach)?

* In Figure 1 category-sensitive electrodes are shown. Based on the description I assume that each highlighted electrode (also listed in the table) is assigned only one category. Thus I would suggest color-coding the categories in Figure one (i.e. separate electrode color for each category).

- * Even though the authors cite a few relevant papers, they do not really acknowledge that in the "conscious perception" literature some relevant works have been performed. Usually these studies use near threshold or ambiguous stimuli to assess the influence of prestimulus activity patterns. The authors may want to check out some prestimulus works focussing on regionally specific alphas oscillations (<https://www.ncbi.nlm.nih.gov/pubmed/24931795>). More importantly in this context may be works using bistable stimuli that can be perceived according to different categories (e.g. <http://www.pnas.org/cgi/pmidlookup?view=long&pmid=18664576>; <https://www.ncbi.nlm.nih.gov/pubmed/21625011>; see also this work linking pre- and post-stim activity <https://www.ncbi.nlm.nih.gov/pubmed/31332019>). Importantly these works already address a central issue raised in this manuscript, namely that endogenous activity patterns are "unspecific".
- * The use of the term "predictive processing" is weak in this manuscript and it is unclear whether the authors really think their work is actually really helping in understanding the "predictive brain" or whether the purpose is to add fancy catchwords. Given the absence of any experimental control of predictive processes, as it stands at the moment I do not think that the argument is strong and these speculations (already in abstract and intro) can be removed without much loss of value. Some speculations in the discussion may be ok. More neutrally, rather than speaking of "predictive" I would suggest using "biased" processing.
- * The choice of prestimulus features appears to me quite "unmotivated" from a conceptual point of view. Why were precisely these features chosen? In general I find it interesting -as shown in Figure 3A- that there seems no precise prestimulus pattern that improves classification, but that the features are more or less en par. Regarding the phase information, I would be surprised if such a sharp peak at ~15 would replicate.
- * I assume that in Figure 2A all prestimulus features were used, right? This would correspond more or less to the "Post+Pre" bar in Figure 3A?
- * p. 18, Stimuli and paradigm description: It is for me unclear how the number "480 independent trials" come about. This means trials which are not a repetition of a previous stimulus? So overall there were ~720 trials?
- * p. 18, task: the 1-back refers to an exact stimulus repetition (e.g. face of Joe shown twice) and not just a category repetition (e.g. face of Joe preceded by face of Jane), right? Perhaps make this explicit.

Minor:

- * In Figure 2A: Add error bars. Also perhaps move Table 1 to Supplementary Materials, as it is essentially showing same information as Figure 2A?
- * in Figure 3 B: add correct color next to "high MI" in legend

Nathan Weisz

Reviewer #2:

Remarks to the Author:

This paper focuses on the influence of prestimulus activity on the accuracy of decoding stimulus category from neural responses to images of objects. The data are obtained from human ECoG recordings across 246 electrodes that are aggregated across 30 epilepsy patients who performed a 1back repetition task on images from five categories (faces, bodies, houses, hammers, words) plus phase-scrambled faces, with most electrodes covering ventral-temporal cortex. The influence of prestimulus activity on category decoding was estimated using a logistic regression approach modified to incorporate prestimulus activity as an additional feature for a 6-way classification of the visually evoked (post-stimulus) activity.

The main result is that including prestimulus activity as a feature improves classification accuracy relative to not including this information. Additional analyses are provided to support the claim that the prestimulus activity also affects behavior (as evidenced by shortened RTs on trials that had lower prestimulus activity) and that these effects are domain-specific, with RT to a given category only being affected by prestimulus activity from electrodes that preferred that category, and with prestimulus activity being shared only by the electrodes that have the same preferred category. The main conclusion is that ongoing neural activity influences the perception of sensory inputs in a 'circuit-specific' (i.e. category-specific) manner, which is interpreted to be more akin to a form of directed internal attention rather than global fluctuations in arousal affecting overall sensory responsiveness.

The topic of this paper -the interaction between endogenously generated and sensory evoked activity- is of interest (as evidenced by the many high impact papers cited), and I think that the ECoG data reported here provide a nice opportunity to study this interaction. The approach taken by the authors seems interesting and valuable, in principle: I liked that they combined a mostly data-driven approach (including many electrodes, and evaluating the influence of multiple data features) with three distinct, testable hypotheses that they clearly made an effort to make explicit for the reader.

However, while I think I understood the overall idea and approach, there were also many aspects of the paper that I found opaque and hard to understand, in particularly the methods. This, combined with the use of relatively abstract language, made it difficult to assess the actual contribution of the paper. The main result and figures felt very removed from the actual data, making it hard to follow the logic of the computational approach and implications of the findings: in the end, it did not become clear to me how exactly the prestimulus activity affects the sensory-evoked neural activity, and which features of the prestimulus activity are important in this context. In some places, I was also worried about selective reporting, as well as potential statistical confounds. I elaborate on these points in more detail below, hoping that the authors will find these useful moving forward.

MAJOR COMMENTS

Language:

* The terms used to describe the hypotheses and motivation are quite abstract, using words like 'tuning', 'circuits', 'neural coding' and 'perceptual quality' that are not clearly defined. At the same time, the terms used in the methods are very specific and jargony, like 'critical discriminant direction/dimension', 'classification boundary', and 'elastic net penalty'. The paper fails to connect these terms to one another in a comprehensible and convincing way. I encourage the authors to drop the fancy words from the framing and stay closer to the data (e.g. just using 'decoding accuracy' instead of 'tuning') and rather direct their energy at taking the reader by the hand, give them with more intuition for what they've actually measured and tested, and by providing more insight in the logic of the overall approach (for example using a figure, see below).

* In some places quite forceful modifiers are used, for example in statements like 'direct evidence', 'without bias' and 'selective modulation': I think one should be careful about using these, and in each case it should be really clear what is meant with these terms and why they are justified here (e.g., why is this evidence 'direct' and other, prior evidence not? Can you ever be sure you are measuring 'without bias'?).

Methods:

* The methods were not sufficiently accessible: the modeling approach needs to be unpacked more. For example, it was not clear to me how the time dimension is handled, is the data in the prestimulus period averaged across time, or are all datapoints used as features? Choices made in preprocessing and model fitting need to be better motivated and explained such that all data transformations going

in the main analysis are clear (see also minor comments below). In the formulas, all the terms need to be properly defined, and they should be presented in the most intuitive order, including motivation for why it was set up that way. For example, it only becomes clear that the 'MI' measure is the main dependent variable of interest somewhere halfway the methods: it would be helpful to state up front that the goal is to estimate this measure.

* I think it would be very helpful to have figure or cartoon with a visualization of the algorithm, the features going into it, that depicts how you go from an example recorded response trace in an electrode to a measure of MI. Another useful addition could be to show a graphic depiction of a low MI and a high MI trial (either from real data or a simulation).

* A major potential confound for the comparison between d' with and without the prestimulus activity seems to me that classification performance is always going to improve by adding more features? This concern can be addressed through cross-validation (fitting the model on half of the data, testing on the other half) and some cross-validation is mentioned on Page 22 but it was not clear to me if this was done for the full two-stage model GLM fits: the sentence seems to suggest this was only done to estimate the regularization parameters, while it would be better if it were done for the entire model.

Results

* I would really like to see some of the actual data, like examples of category responses, their variability across trials and across electrodes, to get a better sense of what these responses look like. Showing the actual responses would also help understand the motivation for the different features better, e.g. why you chose to include both the ERP and the broadband activity.

* Showing more of the actual data may also give the reader more insight into how the magnitude of the responses relates to the decoding accuracy. For example, I wonder if the lack of an RT effect for non-preferred stimuli may be related to the response to the non-preferred stimuli being low magnitude overall, or more variable across trials?

The statistical analysis and results presentation should be improved in a number of ways:

* Error bars are missing in Fig 2A and 3B

* Fig 2B and Fig 3B: why only show lowest and highest quarter of RTs instead of entire distribution? Only comparing these quarters statistically (i.e. omitting the middle of the distribution) needs to be better justified.

* To support the claim that the behavioral effects are 'region-by-stimulus' specific, I think one should test for an interaction effect on the averages provided in Figure 2B, not just showing that there's a difference in one comparison but not another. In fact, I found the comparison of the correlations in the Supplementary Figure actually more convincing than these bar graphs, and would encourage the inclusion of more of those types of results in the main text.

* The RTs were obtained only for repeated trials, so the results reported in Fig 3B are in fact from a smaller subset of the data - it should be more clear how many data points and degrees of freedom went into each plotted average and statistical test, rather than just providing a p-value.

* Instead of bar charts, consider using plotting tools that show the underlying structure and spread of the data, like boxplots, violin plots, or just individual data points.

MINOR COMMENTS

Methods:

* Please provide more detail on the stimuli, e.g. where they were from, their size, approximate degree

of visual angle, whether they were color or grayscale, had backgrounds, if they were processed in any way (e.g. to correct for low-level feature difference in contrast or luminance).

* I was puzzled by the exclusion of the period of the actual stimulus presentation [-100 100] from the analyses. Is this not the point where the actual interaction between endogenous and external inputs interact? In other words, why wouldn't you want to estimate the endogenous activity as close in time to the stimulus as possible?

* The aforementioned sentence on Page 22 about the cross-validation is very vague, 'minimizing the deviance' to what?

* Why do the phase features total to 60? Was this just a random choice, or based on previous work?

* It was not clear to me what the 'critical discriminant dimensions' were, making most of the results unintelligible. For example, the sentence on line 13 "to evaluate the modulation effect along the critical discriminant dimensions..." is really hard to follow even if after having read the Methods section.

Results:

* Fig 1: it would be nice to color code the electrodes by category preference to see to what extent it matches known layout of category-selective regions. You could also consider use size or hue to indicate strength of category selectivity, again just to give the reader a better sense of the data going into the analyses.

* Page 11, lines 1-7: how was VTC defined here?

* Fig 3: I don't really understand how the HFBA only is better than including both; is this a result of cross validation? Would a stepwise version of the regression (where you quantify the gain in predictive power with including more or less features) be a better way to address this question?

* Fig 4: including the autocorrelation on lag 0 in the plot doesn't seem very sensible to me as this always going to be 1: the way it is plotted now this looks as a very salient data point which it is in fact not of interest.

Discussion:

* The main summary statement on Page 15, line 8-11 "prestimulus activity selectively modulates the post-stimulus activity along the critical dimensions that directly link to the tuning of category-selective areas that process the type of stimuli presented" is very hard to follow.

* In the end, I found the claimed stimulus- and circuit specificity of the endogenous activity and its facilitation of detecting a repeated stimulus a little puzzling. The authors frame this in the context of active sensing, and oppose it to a priming effect. But since the participants did not know which category would come up on the next trial, I don't quite follow why it would make sense for the brain to as it were 'pre-engage' a specific circuit (e.g. face network) at the expense of other networks? Or is the idea that they are all engaged in the same way to optimally receive a stimulus, but that this only 'works' for their own preferred stimulus category? In other words, what exactly do the authors think is the potential 'predictive processing mechanism' that is mentioned in the abstract?

Reviewer #3:

Remarks to the Author:

The authors investigated how endogenous brain activity influences neural responses to visual stimuli and related behavior in terms of reaction tasks. Thus, data from 30 patients with intracranial electrodes were classified based on six visual categories. Electrodes selective for visual categories demonstrated evoked responses (i.e., evoked potentials and evoked broad band activity) that were modulated by the pre-stimulus interval. For each response a modulation index was computed, which was based on a two-stage model that assessed how much the classification boundary in a logistic

regression changes based on the pre-stimulus activity. Furthermore, reaction times were monitored to assess how pre-stimulus activity affects behavior.

- What are the major claims of the paper?

The results of this study demonstrate three main findings:

1. endogenous activity influences trial-by-trial neural tuning in a stimulus and circuit specific manner;
2. the same aspect of the endogenous activity that modulates neural tuning also facilitates behavior;
3. fluctuations of this aspect of endogenous activity are not globally correlated, rather these fluctuations are only correlated across networks sensitive to the same stimulus category.

• Are the claims novel? If not, please identify the major papers that compromise novelty
Yes, the authors demonstrated the differences to previous studies, including recent publications.

- Will the paper be of interest to others in the field?

Yes, numerous of papers addressed the effect of endogenous brain activity and these findings will be of interest for further studies.

- Will the paper influence thinking in the field?

Not really. The paper supports previous findings, for example that endogenous brain activity biases or influences responses to upcoming stimuli. However, the current study extends these findings by its demonstration that both responses and behavior are affected by the same mechanism.

- Are the claims convincing? If not, what further evidence is needed?

Partly. The major claims above seemed to be adequately proven. Sometimes the authors claim that perception of sensory input is influenced by the present neural state. However, the perception itself is not directly addressed in this study, only neural responses and behavior. Neither changes in percepts nor misclassifications related to wrongly perceived stimuli were mentioned. Therefore, authors should be careful provide more evidence that perception changed or remove that claims.

Page 3 Line 2: "Endogenous brain states influence both neural responses and perception of stimuli." It is not clear how the perception of the subjects was influenced. "Perception" could be replaced by "behavior", which was clearly shown in the study. Based on the manuscript only the behavior in the reaction task and the neural responses changed. How can the authors infer from the study that a subject's perception was influenced?

Page 2 Line 10: "modulated behavior" is a very strong and general inference, and requires more explanation to better understand maybe by adding "in terms of reaction times" or similar.

- Are the claims appropriately discussed in the context of previous literature?

Yes, the authors demonstrated the differences to previous studies, including recent publications.

- If the manuscript is unacceptable in its present form, does the study seem sufficiently promising that the authors should be encouraged to consider a resubmission in the future?

Yes.

- Is the manuscript clearly written? If not, how could it be made more accessible?

In general, yes.

RT is assumed to be reaction time, but was not defined as such.

Page 18 Line 16/17: The following sentence occurs again in line 22 and should be removed: "At random, 1/3 of the time an image would be repeated, which yielded 480 independent trials in each session."

- Could the manuscript be shortened to aid communication of the most important findings?

No.

- Have the authors done themselves justice without overselling their claims?

Yes.

- Have they been fair in their treatment of previous literature?

Yes.

- Have they provided sufficient methodological detail that the experiments could be reproduced?

Page 18 Line 21: Was the visual angle the same for all images in the experiment or only the fixation cross?

Page 18 Line 24: Is the reaction time the period from stimulus onset until the button is pressed in the 1- back task?

Page 18 Line 22: Seems that RT analysis are based on 120 trials rather than 480. If so, were the trials ranked by MI only those with RT or all of them and how many RT trials were included in the test?

Page 19 Line 13: What was the reference period for z-scoring the data? Was it based on the pre-stimulus intervals or on the whole data set?

Page 20: The electrode selection procedure is not fully explained. How was the separation between training and test data done? Were true positive and false positive rates based on cross-validated results?

Page 21 Line 11: "critical discriminant dimensions (denoted as β_{evk^*}) should be added to explain all variables. Also, X_{evk} should be explained as the input features. The cross-entropy loss for logistic regression is not clear. Typically the cost function is $J(\beta) = \frac{1}{m} (-yT \log(\frac{1}{1+e^{(\beta x)}}) - (1-y)T(\log(1 - \frac{1}{1+e^{(\beta x)}})))$), can the authors explain the cost function in more detail?

Page 21/22: Again, the separation between training and test data is not clear. Please explain how cross-validation was performed and how overfitting was avoided.

Page 23 Line 15: The selected 2cm rule for cross-correlation seems a bit strict. What was the electrode spacing of the intracranial grids and stereo EEG? How many tested pairs remained after applying the rule?

Page 24 Line 6: The parentheses seem to be redundant in the equation.

- Is the statistical analysis of the data sound?

Yes, if the classification of the data was done on unseen test data.

Page 13 Line 4/5: "To evaluate the contribution of different aspects of the endogenous features, the same model was applied using different subsets of the pre-stimulus features." Repeated tests with mixed features require a correction, in the most conservative case a Bonferroni correction.

- Should the authors be asked to provide further data or methodological information to help others replicate their work? (Such data might include source code for modelling studies, detailed protocols or mathematical derivations).

Yes, the methods have to be explained and clarified.

- Are there any special ethical concerns arising from the use of animals or human subjects?

Data were recorded from patients who underwent epilepsy monitoring and thus, required intracranial electrodes for clinical reasons. The study protocol was approved and the patients were informed.

Reviewers' comments:

Reviewer #1 (Remarks to the Author):

In their manuscript Li et al. address the interesting question to what extent "endogenous" (rather prestimulus) activity shape visual category specific neural responses and whether these - traditionally usually neglected- neural pattern are also behaviorally relevant. To this end they use iEEG data recorded from 30 epileptic patients and apply a multivariate regression approach to first identify classification based on post-stimulus data and then adding -in a second step- prestimulus data in order to test whether classification accuracy (d-prime measure) is improved. From the latter analysis they are able to quantify the extent to which prestimulus information changes model weights and use this measure (~Modulation Index) to follow up with further analysis on a single-trial level. Among other aspects they are able to show that the MI measures correlate between electrodes with similar category sensitivity, whereas they are virtually uncorrelated between electrodes that are sensitive to different categories. Also the authors claim that lower prestimulus activity (~lower LFP and gamma band power) go along facilitatory / biasing effects.

Overall there is a lot that I like about this study, even though I admit to struggle with the details of the methodological approach and will freely admit that I probably cannot judge whether all is done in a satisfactory manner. However, also given the chosen "general interest" format of the journal, I think the authors should make a stronger effort in making their approach more accessible in general terms at strategically well-chosen points (outside of the methods section). Furthermore, I think that central claims -at this stage- are not well backed up: especially the aspect that the (prestimulus) results support an improved neural tuning at a perceptual, rather than at a decision level. Thus for the moment the manuscript requires significant improvements before publication can be considered.

We greatly appreciate the reviewer's constructive and positive comments about our work and feel that the work has been improved by making the approach and results more approachable.

Issues:

* The most crucial issue in my opinion is that the authors intend to communicate that their prestimulus patterns bias "neural tuning" supporting perceptual processing. This is not clearly shown and the effects (neurally, but also the RT effects) could equally well be biasing decision making processes. In fact many motor and prefrontal electrodes appear as category sensitive (Figure 1). While I do not exclude the possibility that they may be relevant for category perception, the fact that the task involved (speeded) motor responses does not help in disambiguating these different contributing influences.

We agree that the results do not fully disentangle prestimulus activity influencing decision processes versus perception per se. That said, the results hold when only VTC electrodes are included (excluding motor and prefrontal electrodes; see the new supplement table S2). Previous studies have linked the multivariate representation of the activity in VTC to the behavioral perceptual representation (e.g. Charest et al. PNAS 2014), which supports the idea that we are examining the effects on perceptual processing.

We now expand this discussion on page 18 and acknowledge that our work cannot fully exclude the possibility that what we see is related to decision, rather than perceptual, processes.

Two small points to note: 1) The number of prefrontal, motor, and parietal electrodes that are category selective are a small proportion of the total number of electrodes across participants in these regions and a much smaller proportion than in VTC. These were primarily included to show that the results may generalize outside of VTC and in fact in early stages of this work we only looked at VTC electrodes with similar results (now presented as Table S2 in supplement). 2) A large majority of the frontal electrodes are selective for words, which may relate to language processes in these regions.

* Regarding the interpretation that "low activity" biases neural tuning and behavior in category specific manners, I am not sure to what extent this claim can be made in a strong manner. This pattern could in principle be driven by regionally specific power desynchronizations (which can be regionally extremely specific; <https://www.ncbi.nlm.nih.gov/pubmed/31291043>), that can go along with enhanced frequency of action potentials (<https://www.ncbi.nlm.nih.gov/pubmed/22084106>). Thus, the results could in principle support the quite opposite conclusion.

We thank the reviewer for pointing out these references. The reviewer is correct that in principle the results could be driven either by reduced neuronal activity or desynchronization and our results cannot detangle these two possibilities as we do not have single unit measurements. However, given the results from the Dragoi group showing similar improvements of tuning and behavior in early visual cortex, albeit without stimulus and circuit level specificity, we suggest it may be more likely that activity is reduced. We have now noted on page 19 (lines 16-18) the alternative possibility that the reduced mean and variance of the prestimulus activity could be a result of desynchronization and reference the articles that the reviewer suggests. We also note that given the results from the Dragoi group, the results seen here may instead suggest a generalization of the mechanisms that they have described at the neuronal level.

* As stated above, I will not pretend to understand all details of the methodological approach. However with solid M/EEG background I would consider myself a representative reader that is interested in the general topic. I would really welcome a better effort to take the reader by the hand to understand the critical elements. This would include some helpful "fool-proof" comments here and there, but also I think a Figure (e.g. added to Figure 1) visualizing some aspects (e.g. how the MI is derived, which is critical for several conclusions). The methods descriptions makes me assume that regression models were computed for every trial and electrode separately, is this correct (also asking because this is not a mainstream approach)?

As mentioned in the response to the editor above, we have expanded our description of the methods. In particular, we have also added a detailed Algorithm 1 and supplement materials to explain how the model fitting works in details. We have also included a cartoon of the method and how MI is derived in Figure 1.

* In Figure 1 category-sensitive electrodes are shown. Based on the description I assume that each highlighted electrode (also listed in the table) is assigned only one category. Thus I would

suggest color-coding the categories in Figure one (i.e. separate electrode color for each category).

We thank the reviewer for this useful suggestion. We have now color coded the electrodes by category selectivity and used size to indicate strength of category selectivity.

* Even though the authors cite a few relevant papers, they do not really acknowledge that in the "conscious perception" literature some relevant works have been performed. Usually these studies use near threshold or ambiguous stimuli to assess the influence of prestimulus activity patterns. The authors may want to check out some prestimulus works focussing on regionally specific alpha-like oscillations (<https://www.ncbi.nlm.nih.gov/pubmed/24931795>). More importantly in this context may be works using bistable stimuli that can be perceived according to different categories (e.g. <http://www.pnas.org/cgi/pmidlookup?view=long&pmid=18664576>; <https://www.ncbi.nlm.nih.gov/pubmed/21625011>; see also this work linking pre- and post-stim activity <https://www.ncbi.nlm.nih.gov/pubmed/31332019>). Importantly these works already address a central issue raised in this manuscript, namely that endogenous activity patterns are "unspecific".

We thank the reviewer for pointing this out. We have now included these citations in the introduction (page 3 line 9).

* The use of the term "predictive processing" is weak in this manuscript and it is unclear whether the authors really think their work is actually really helping in understanding the "predictive brain" or whether the purpose is to add fancy catchwords. Given the absence of any experimental control of predictive processes, as it stands at the moment I do not think that the argument is strong and these speculations (already in abstract and intro) can be removed without much loss of value. Some speculations in the discussion may be ok. More neutrally, rather than speaking of "predictive" I would suggest using "biased" processing.

We appreciate this comment and agree with the reviewer that perhaps the link to "predictive processing" was previously pushed too much. The idea that we were trying to convey is that for the hypothesis that endogenous activity is a signature of predictive coding to be correct (e.g. Fiser TICS 2010), stimulus and circuit specific modulation of discrimination is a necessary (though not sufficient) feature of endogenous activity. We now make this clearer on page 20 (lines 6-11) and limit the discussion of predictive processing throughout as our results do not demonstrate that the prestimulus activity is a reflection of predictive processing, but rather that our results provide evidence for a property that is necessary (though not sufficient), and is therefore consistent with, predictive processing.

* The choice of prestimulus features appears to me quite "unmotivated" from a conceptual point of view. Why were precisely these features chosen? In general I find it interesting -as shown in Figure 3A- that there seems no precise prestimulus pattern that improves classification, but that the features are more or less en par. Regarding the phase information, I would be surprised if such a sharp peak at ~15 would replicate.

There is no clear consensus on what aspects of the signal should be used in iEEG studies. We are generally of the opinion that the straightforward potentials measured from the electrodes is the most unbiased signal to be used, which is why we generally always include the single trial potentials in our analyses. In addition, the broadband high-frequency activity is widely used in iEEG studies and has previously been shown to be the aspect of the signal best correlated with the population firing rate, so we included this as well as a separate feature. Note that the stBHA is in the stP, but is relatively hidden because of the $1/f$ bias in the power spectrum of neural activity. Furthermore, both stBHA and stP carry critical information about the visual category and also have been shown to capture correlated, but also complementary aspects of the signal (Miller et al. PloS Comp Bio 2016). Finally, the oscillatory phases are used because previous studies have indicated the potential influence of prestimulus phase and perception (e.g. Busch et al J Neurosci 2009, Dugué et al J Neurosci 2011). Moreover, we used a sparse regression method (L1 norm regularization), and the redundant or non-informative features would be ignored automatically. Therefore, it is better to include more potentially informative features into the model. One likely interpretation of the results in Fig 3A is that these features may be correlated, which is why across electrodes and subjects none are particularly favored over others.

Also, we make no assertions that the sharp 15 Hz peak would replicate. Rather we emphasize that the frequency region from 10-25 Hz was above chance and have made this clearer on page 15 (lines 24-25).

* I assume that in Figure 2A all prestimulus features were used, right? This would correspond more or less to the "Post+Pre" bar in Figure 3A?

Yes, this is correct.

* p. 18, Stimuli and paradigm description: It is for me unclear how the number "480 independent trials" come about. This means trials which are not a repetition of a previous stimulus? So overall there were ~720 trials?

There are 180 individual images and they are presented twice, which yields 360 image presentations. And 1/3 of the time, an image would be repeated, thus $180 * 2 * (4/3) = 480$ total trials. We have now changed "480 independent trials" into "480 total trials" to avoid confusion.

* p. 18, task: the 1-back refers to an exact stimulus repetition (e.g. face of Joe shown twice) and not just a category repetition (e.g. face of Joe preceded by face of Jane), right? Perhaps make this explicit.

Yes. We have now made it clear that it is referring to "an exact image was repeated".

Minor:

* In Figure 2A: Add error bars. Also perhaps move Table 1 to Supplementary Materials, as it is essentially showing same information as Figure 2A?

* in Figure 3 B: add correct color next to "high MI" in legend

We thank the reviewer for pointing these out. We have revised the figures accordingly.

Nathan Weisz

Reviewer #2 (Remarks to the Author):

This paper focuses on the influence of prestimulus activity on the accuracy of decoding stimulus category from neural responses to images of objects. The data are obtained from human ECoG recordings across 246 electrodes that are aggregated across 30 epilepsy patients who performed a 1back repetition task on images from five categories (faces, bodies, houses, hammers, words) plus phase-scrambled faces, with most electrodes covering ventral-temporal cortex. The influence of prestimulus activity on category decoding was estimated using a logistic regression approach modified to incorporate prestimulus activity as an additional feature for a 6-way classification of the visually evoked (post-stimulus) activity.

The main result is that including prestimulus activity as a feature improves classification accuracy relative to not including this information. Additional analyses are provided to support the claim that the prestimulus activity also affects behavior (as evidenced by shortened RTs on trials that had lower prestimulus activity) and that these effects are domain-specific, with RT to a given category only being affected by prestimulus activity from electrodes that preferred that category, and with prestimulus activity being shared only by the electrodes that have the same preferred category. The main conclusion is that ongoing neural activity influences the perception of sensory inputs in a 'circuit-specific' (i.e. category-specific) manner, which is interpreted to be more akin to a form of directed internal attention rather than global fluctuations in arousal affecting overall sensory responsiveness.

The topic of this paper -the interaction between endogenously generated and sensory evoked activity- is of interest (as evidenced by the many high impact papers cited), and I think that the ECoG data reported here provide a nice opportunity to study this interaction. The approach taken by the authors seems interesting and valuable, in principle: I liked that they combined a mostly data-driven approach (including many electrodes, and evaluating the influence of multiple data features) with three distinct, testable hypotheses that they clearly made an effort to make explicit for the reader.

However, while I think I understood the overall idea and approach, there were also many aspects of the paper that I found opaque and hard to understand, in particular the methods. This, combined with the use of relatively abstract language, made it difficult to assess the actual contribution of the paper. The main result and figures felt very removed from the actual data, making it hard to follow the logic of the computational approach and implications of the findings: in the end, it did not become clear to me how exactly the prestimulus activity affects the sensory-evoked neural activity, and which features of the prestimulus activity are important in this context. In some places, I was also worried about selective reporting, as well as potential statistical confounds. I elaborate on these points in more detail below, hoping that the authors will find these useful moving forward.

We thank the reviewer for his/her positive remarks about our work and constructive feedback. We feel the work has been improved by addressing them.

MAJOR COMMENTS

Language:

* The terms used to describe the hypotheses and motivation are quite abstract, using words like ‘tuning’, ‘circuits’, ‘neural coding’ and ‘perceptual quality’ that are not clearly defined. At the same time, the terms used in the methods are very specific and jargony, like ‘critical discriminant direction/dimension’, ‘classification boundary’, and ‘elastic net penalty’. The paper fails to connect these terms to one another in a comprehensible and convincing way. I encourage the authors to drop the fancy words from the framing and stay closer to the data (e.g. just using ‘decoding accuracy’ instead of ‘tuning’) and rather direct their energy at taking the reader by the hand, give them with more intuition for what they’ve actually measured and tested, and by providing more insight in the logic of the overall approach (for example using a figure, see below).

Thank you for this comment. We agree that we did not well connect the theoretical ideas with the concrete instantiations of these ideas in the methods and results. We have now tried to be clearer, for example using “tuning” when talking about conceptual ideas about the interpretation and motivation of the work and using “classification accuracy” when reporting results or discussing methods. We also try to make the connection between these ideas and their instantiation clearer.

* In some places quite forceful modifiers are used, for example in statements like ‘direct evidence’, ‘without bias’ and ‘selective modulation’: I think one should be careful about using these, and in each case it should be really clear what is meant with these terms and why they are justified here (e.g., why is this evidence ‘direct’ and other, prior evidence not? Can you ever be sure you are measuring ‘without bias’?).

We have tried to clarify these issues throughout. For example when we say “Here we presented direct evidence that the two processes can be attributed to the same aspects of pre-stimulus activity in the same local category-sensitive circuit.” we are trying to draw a distinction to previous studies that have shown a relationship between prestimulus activity and poststimulus response and other studies that have shown a relationship between prestimulus activity and behavior, but have not demonstrated that it is the same aspect of prestimulus activity can be attributed to the relationship to both. We have taken out the word “direct” as we understand what “direct” was referring to is unclear and may be an unnecessary modifier. In general we have removed the modifiers the reviewer suggests.

Methods:

* The methods were not sufficiently accessible: the modeling approach needs to be unpacked more. For example, it was not clear to me how the time dimension is handled, is the data in the prestimulus period averaged across time, or are all datapoints used as features? Choices made in preprocessing and model fitting need to be better motivated and explained such that all data transformations going in the main analysis are clear (see also minor comments below). In the formulas, all the terms need to be properly defined, and they should be presented in the most intuitive order, including motivation for why it was set up that way. For example, it only becomes

clear that the 'MI' measure is the main dependent variable of interest somewhere halfway the methods: it would be helpful to state up front that the goal is to estimate this measure.

We have now added additional information about the feature dimensions in the model on page 30. All individual time points in the pre-stimulus and post-stimulus periods are used as features in the model (e.g. not averaged, but each timepoint was used separately as a feature in the multivariate model). We have also added a table to clarify the flow of the methods (algorithm box in the methods section) and a qualitative cartoon of the method as Figure 1C.

* I think it would be very helpful to have figure or cartoon with a visualization of the algorithm, the features going into it, that depicts how you go from an example recorded response trace in an electrode to a measure of MI. Another useful addition could be to show a graphic depiction of a low MI and a high MI trial (either from real data or a simulation).

We have also added pseudo code of the algorithm (Algorithm 1, page 30) and supplement materials to explain how the model fitting works in details.

* A major potential confound for the comparison between d' with and without the prestimulus activity seems to me that classification performance is always going to improve by adding more features? This concern can be addressed through cross-validation (fitting the model on half of the data, testing on the other half) and some cross-validation is mentioned on Page 22 but it was not clear to me if this was done for the full two-stage model GLM fits: the sentence seems to suggest this was only done to estimate the regularization parameters, while it would be better if it were done for the entire model.

We thank the reviewer for pointing this out as the previous version of the manuscript was unclear. The cross-validation was done for both the single stage model and full two-stage model of the GLM fits. We have now added additional section in the Methods part explaining that cross-validation was performed to select hyperparameters and avoid overfitting (pages 27-28).

Results

* I would really like to see some of the actual data, like examples of category responses, their variability across trials and across electrodes, to get a better sense of what these responses look like. Showing the actual responses would also help understand the motivation for the different features better, e.g. why you chose to include both the ERP and the broadband activity.

We thank the reviewer for this useful suggestion. We have now included an example of category-selective electrodes in supplement Fig S1, demonstrating the selectivity in both the ERP and the broadband activity, as well as the variability across trials. As we note in our response to Reviewer 1, using the stP, the stBHA, and the phase features was an a priori decision made for theoretical reasons based on the literature.

* Showing more of the actual data may also give the reader more insight into how the

magnitude of the responses relates to the decoding accuracy. For example, I wonder if the lack of an RT effect for non-preferred stimuli may be related to the response to the non-preferred stimuli being low magnitude overall, or more variable across trials?

We concur with the reviewer's assessment that the response to non-preferred stimuli is of low magnitude overall. However, we are not examining the correlation between the stimulus response and behavior, but rather the pre-stimulus MI and behavior. Thus one key is the fact that there is little correlation of the MI between preferred and non-preferred electrodes. This point is critical to our message in our opinion as it further supports the specificity of the effects. EG if the effects we report were a result of a global, non-specific process such as arousal, the MI should correlate with behavior even for non-preferred stimuli.

The statistical analysis and results presentation should be improved in a number of ways:

* Error bars are missing in Fig 2A and 3B

Figure 2 and 3 have been revised accordingly to better demonstrate the entire distribution.

* Fig 2B and Fig 3B: why only show lowest and highest quarter of RTs instead of entire distribution? Only comparing these quarters statistically (i.e. omitting the middle of the distribution) needs to be better justified.

The main reason for doing this is to show the difference in RT, since the middle quarters may not be largely separated. The correlation of the entire distribution was shown in Figure S1 in the original submission and this correlation was statistically significant. We have now also included the averaged trial-by-trial correlation between RT and MI in Figure 2C, which includes the middle of the distribution.

* To support the claim that the behavioral effects are 'region-by-stimulus' specific, I think one should test for an interaction effect on the averages provided in Figure 2B, not just showing that there's a difference in one comparison but not another. In fact, I found the comparison of the correlations in the Supplementary Figure actually more convincing than these bar graphs, and would encourage the inclusion of more of those types of results in the main text.

We thank the reviewer for this useful suggestion. We have now included the comparisons of the averaged trial-by-trial correlation between RT and MI into Figure 2.

* The RTs were obtained only for repeated trials, so the results reported in Fig 3B are in fact from a smaller subset of the data - it should be more clear how many data points and degrees of freedom went into each plotted average and statistical test, rather than just providing a p-value.

It is true that the RTs were obtained only for repeated trials. Fig 3B is actually ranking all trials in the analysis. The reviewer might refer to Fig 2B, which indeed only takes into account 1/4 of the total trials (~120 per block) that were repeated. We have made it clear in the text and figure legend how many data points went into each plot.

* Instead of bar charts, consider using plotting tools that show the underlying structure and spread of the data, like boxplots, violin plots, or just individual data points.

We thank the reviewer for this useful suggestion. We have now revised Figure 2 and Fig 3 to show the distribution of individual datapoints. Specifically, we use beeswarm plots in Fig 2 and Fig 3 to show the distributions and to demonstrate the different numbers of samples that went into different cases.

MINOR COMMENTS

Methods:

* Please provide more detail on the stimuli, e.g. where they were from, their size, approximate degree of visual angle, whether they were color or grayscale, had backgrounds, if they were processed in any way (e.g. to correct for low-level feature difference in contrast or luminance).

The images are all of the same size and $\sim 10 \times 10$ degree of visual angle. They were all presented in grayscale with white background (see Fig 1A for examples). There was no explicit corrections for lower level feature difference in contrast or luminance.

* I was puzzled by the exclusion of the period of the actual stimulus presentation $[-100 \ 100]$ from the analyses. Is this not the point where the actual interaction between endogenous and external inputs interact? In other words, why wouldn't you want to estimate the endogenous activity as close in time to the stimulus as possible?

We agree that this period is the place where the maximal interaction might occur, but this is also the period when there is potential signal leakage. Specifically, there is potential signal leakage caused by the low-pass and band-pass filters, including hardware filters. Thus, we exclude this period out of an abundance of caution to ensure there was no spillover of prestimulus activity into the activity used as the stimulus response (and vice versa) in a conservative manner. We now mention this on pages 23-24 (lines 19-23, 1-3) of the methods.

* The aforementioned sentence on Page 22 about the cross-validation is very vague, 'minimizing the deviance' to what?

The deviance refers to the loss function of the logistic regression (i.e. cross-entropy loss). We have now added additional section in the Methods part explaining how cross-validation was performed (pages 27-28).

* Why do the phase features total to 60? Was this just a random choice, or based on previous work?

We included phase values from 0 to 150 Hz, and we arbitrarily chose 2.5 hz as the step size, therefore we ended up with 60 phase values for the entire frequency range (every 2.5 hz).

* It was not clear to me what the ‘critical discriminant dimensions’ were, making most of the results unintelligible. For example, the sentence on line 13 “to evaluate the modulation effect along the critical discriminant dimensions...” is really hard to follow even if after having read the Methods section.

It is referring to the dimension that best separates the post-stimulus evoked response between categories, i.e. β_{evk}^* . Essentially all linear classifiers in the end can be thought of as finding a “discriminant dimension” (perpendicular to the hyperplane boundary) and then projecting the test data onto this hyperplane and classifying the category the data came from based on its location in this discriminant dimension relative to the boundary point (where the dimension crosses the hyperplane). We have tried to clarify this in the text and use the cartoon in Fig 1C as a demonstration. Also, we have reduced the use of the jargon “discriminant dimension” as much as possible.

Results:

* Fig 1: it would be nice to color code the electrodes by category preference to see to what extent it matches known layout of category-selective regions. You could also consider use size or hue to indicate strength of category selectivity, again just to give the reader a better sense of the data going into the analyses.

We thank the reviewer for this useful suggestion. We have now color coded the electrodes by category selectivity and used size to indicate strength of category selectivity.

* Page 11, lines 1-7: how was VTC defined here?

VTC (ventral temporal cortex) refers to the ventral surface of the temporal lobe, including mostly inferior temporal gyrus, fusiform gyrus, parahippocampal gyrus. We have added additional description of the definition.

* Fig 3: I don’t really understand how the HFBA only is better than including both; is this a result of cross validation? Would a stepwise version of the regression (where you quantify the gain in predictive power with including more or less features) be a better way to address this question?

It is a result of cross-validation and although it appears to be slightly higher there is no significant difference between HFBA and the others. That said, theoretically removing uninformative features can improve classification accuracy. The inclusion of more and less features can be seen for the Post+Pre (which includes all features, then moving to the stP+stBHA, stBHA+Ph, stP+Ph, then moving to stP, stBHA, and Ph alone.

* Fig 4: including the autocorrelation on lag 0 in the plot doesn’t seem very sensible to me as this always going to be 1: the way it is plotted now this looks as a very salient data point which it is in fact not of interest.

We thank the reviewer for this suggestion. We have revised Figure 4 to make it start from lag 1.

Discussion:

* The main summary statement on Page 15, line 8-11 “prestimulus activity selectively modulates the post-stimulus activity along the critical dimensions that directly link to the tuning of category-selective areas that process the type of stimuli presented” is very hard to follow.

We have updated this sentence to: “The results here demonstrate that pre-stimulus activity selectively modulates the post-stimulus activity along the critical category selective dimensions in these areas.”

* In the end, I found the claimed stimulus- and circuit specificity of the endogenous activity and its facilitation of detecting a repeated stimulus a little puzzling. The authors frame this in the context of active sensing, and oppose it to a priming effect. But since the participants did not know which category would come up on the next trial, I don't quite follow why it would make sense for the brain to as it were 'pre-engage' a specific circuit (e.g. face network) at the expense of other networks? Or is the idea that they are all engaged in the same way to optimally receive a stimulus, but that this only 'works' for their own preferred stimulus category? In other words, what exactly do the authors think is the potential 'predictive processing mechanism' that is mentioned in the abstract?

We suggest that our results do not support the idea of a priming-like effect of pre-activation because that kind of pre-activation should result in increased prestimulus activity being associated with improved behavior and classification, rather than decreased activity (both mean and variance) as we report. As the reviewer described, the results suggest something closer to “pre-engagement” to optimally receive input, though the “pre-engagement” is not a priming-like “pre-activation.”

As for why this is seen despite not providing any behavioral advantage given that there is random presentation, that is not entirely clear. As we note on pages 18-19:

“Given the random stimulus presentation in the present study, facilitating one stimulus over another on a trial-by-trial basis does not provide a behavioral advantage. Therefore, it is unclear if the endogenous activity seen here reflects stochastic dynamics in brain circuits, such as fluctuations of neurotransmitter levels⁴⁷, or strategic processes, such as fluctuations in stimulus-specific attention or preference⁴⁸, that may reflect pattern detection and strategies primates adopt even when stimuli are presented randomly⁴⁹.”

Thus, we suggest that it is due to some combination of stochastic local processes and/or the primate tendency to try to find patterns/strategies even in noise. It may be that when there is behavioral advantage, such as in natural settings where the scene is often familiar, the effects seen here are enhanced. We now try to make this logic a bit clearer on pages 18-19.

Reviewer #3 (Remarks to the Author):

We thank the reviewer for his/her thorough read of our work and important comments. We feel addressing these points have improved the work.

The authors investigated how endogenous brain activity influences neural responses to visual stimuli and related behavior in terms of reaction tasks. Thus, data from 30 patients with intracranial electrodes were classified based on six visual categories. Electrodes selective for visual categories demonstrated evoked responses (i.e., evoked potentials and evoked broad band activity) that were modulated by the pre-stimulus interval. For each response a modulation index was computed, which was based on a two-stage model that assessed how much the classification boundary in a logistic regression changes based on the pre-stimulus activity. Furthermore, reaction times were monitored to assess how pre-stimulus activity affects behavior.

• What are the major claims of the paper?

The results of this study demonstrate three main findings:

1. endogenous activity influences trial-by-trial neural tuning in a stimulus and circuit specific manner;
2. the same aspect of the endogenous activity that modulates neural tuning also facilitates behavior;
3. fluctuations of this aspect of endogenous activity are not globally correlated, rather these fluctuations are only correlated across networks sensitive to the same stimulus category.

• Are the claims novel? If not, please identify the major papers that compromise novelty

Yes, the authors demonstrated the differences to previous studies, including recent publications.

• Will the paper be of interest to others in the field?

Yes, numerous of papers addressed the effect of endogenous brain activity and these findings will be of interest for further studies.

• Will the paper influence thinking in the field?

Not really. The paper supports previous findings, for example that endogenous brain activity biases or influences responses to upcoming stimuli. However, the current study extends these findings by its demonstration that both responses and behavior are affected by the same mechanism.

• Are the claims convincing? If not, what further evidence is needed?

Partly. The major claims above seemed to be adequately proven. Sometimes the authors claim that perception of sensory input is influenced by the present neural state. However, the perception itself is not directly addressed in this study, only neural responses and behavior. Neither changes in percepts nor misclassifications related to wrongly perceived stimuli were mentioned. Therefore, authors should be careful provide more evidence that perception changed or remove that claims.

Page 3 Line 2: “Endogenous brain states influence both neural responses and perception of stimuli.” It is not clear how the perception of the subjects was influenced. “Perception” could be replaced by “behavior”, which was clearly shown in the study. Based on the manuscript only the behavior in the reaction task and the neural responses changed. How can the authors infer from the study that a subject’s perception was influenced?

We thank the reviewer for raising this question. As we discuss in the first response to reviewer one, we agree that it is hard to draw too strong an inference that perception was influenced. That said, the results hold when only VTC electrodes are included.

Previous studies have linked the multivariate representation of the activity in VTC to behavior (e.g. Charest et al. PNAS 2014), which supports the idea that perceptual processes are influenced. We now expand this discussion on page 18 and acknowledge that our work cannot fully exclude the possibility that what we see is related to other aspects of behavior rather than a strictly perceptual process.

Page 2 Line 10: “modulated behavior” is a very strong and general inference, and requires more explanation to better understand maybe by adding “in terms of reaction times” or similar.

We thank the reviewer for this suggestion. We have added “in terms of reaction times” to the sentence to make it clear.

• Are the claims appropriately discussed in the context of previous literature?

Yes, the authors demonstrated the differences to previous studies, including recent publications.

• If the manuscript is unacceptable in its present form, does the study seem sufficiently promising that the authors should be encouraged to consider a resubmission in the future?

Yes.

• Is the manuscript clearly written? If not, how could it be made more accessible?

In general, yes.

RT is assumed to be reaction time, but was not defined as such.

We thank the reviewer for pointing it out. We have now added the definition of RT in Results (page 5 lines 7-9) and Methods sections (page 22, lines 23-24).

Page 18 Line 16/17: The following sentence occurs again in line 22 and should be removed: “At random, 1/3 of the time an image would be repeated, which yielded 480 independent trials in each session.”

We thank the reviewer for pointing it out. This sentence has been removed.

• Could the manuscript be shortened to aid communication of the most important findings?

No.

• Have the authors done themselves justice without overselling their claims?

Yes.

• Have they been fair in their treatment of previous literature?

Yes.

• Have they provided sufficient methodological detail that the experiments could be reproduced?

Page 18 Line 21: Was the visual angle the same for all images in the experiment or only the fixation cross?

We thank the reviewer for pointing out the ambiguous description. The visual angle is the same for all images. We have now made it clear in the text on page 22 (lines 20-21).

Page 18 Line 24: Is the reaction time the period from stimulus onset until the button is pressed in the 1- back task?

Yes, it is. We thank the reviewer for pointing out the ambiguity in the description. We have now made this definition explicit in the text on page 21 (lines 23-24).

Page 18 Line 22: Seems that RT analysis are based on 120 trials rather than 480. If so, were the trials ranked by MI only those with RT or all of them and how many RT trials were included in the test?

For the RT analysis, only the 120 trials with RT in each block were included in the analysis. And these trials were ranked by MI only with the trials that have RT. All these 120 trials with RT were included in the test. We have added additional explanation about how the MI values were computed in cross-validation in the Methods part on pages 27-28.

Page 19 Line 13: What was the reference period for z-scoring the data? Was it based on the pre-stimulus intervals or on the whole data set?

The z-scoring was performed based on the mean and standard deviation of the brief non-task period between each block. This is now clarified on page 22 (lines 16-17).

Page 20: The electrode selection procedure is not fully explained. How was the separation between training and test data done? Were true positive and false positive rates based on cross-validated results?

We thank the reviewer for pointing this out. The electrode selection was performed based on similar 5-fold cross-validation procedure as the other classification analysis in this study. We have added this information on pages 27-28.

Page 21 Line 11: "critical discriminant dimensions (denoted as beta_evk*) should be added to explain all variables. Also, X_evk should be explained as the input features. The cross-entropy loss for logistic regression is not clear. Typically the cost function is $J(\beta) = 1/m (-y^T \log(1/(1+e^{(\beta x)})) - (1-y)^T \log(1/(1+e^{(\beta x)})))$, can the authors explain the cost function in more detail?

We thank the reviewer for pointing this out. We have added the explanation of all variable notations to the Methods part.

In the loss function is actually referring to $\ell(z) = -y^T z + \mathbf{1}^T \log(1 + \exp(z))$, where $z = X\beta$. To avoid confusion, the loss function is now written as $\ell(\beta) = -y^T x^T \beta + \mathbf{1}^T \log(1 + \exp(x^T \beta))$. It is easy to see that this is equivalent to the original form that the reviewer was referring to:

$$\begin{aligned}\ell(\beta) &= \frac{1}{m} \left(-y^T \log \left(\frac{1}{1 + \exp(-x^T \beta)} \right) - (1 - y)^T \log \left(1 - \frac{1}{1 + \exp(-x^T \beta)} \right) \right) \\ &= \frac{1}{m} \left(-y^T \log \left(\frac{\exp(x^T \beta)}{1 + \exp(x^T \beta)} \right) - (1 - y)^T \log \left(\frac{1}{1 + \exp(x^T \beta)} \right) \right) \\ &= \frac{1}{m} \left(-y^T x^T \beta - \mathbf{1}^T \log \left(\frac{1}{1 + \exp(x^T \beta)} \right) \right) \\ &= \frac{1}{m} (x^T \beta + \mathbf{1}^T \log(1 + \exp(x^T \beta)))\end{aligned}$$

For simplicity of notation, we use the last equation, and since $1/m$ does not affect the optimization, we ignore this factor. This form of loss function can be found in Hastie et al. (2009) *The Elements of Statistical Learning*.

Page 21/22: Again, the separation between training and test data is not clear. Please explain how cross-validation was performed and how overfitting was avoided.

We thank the reviewer for pointing this out. We have now added additional section in the Methods part explaining how cross-validation was performed to select hyperparameters and avoid overfitting.

Page 23 Line 15: The selected 2cm rule for cross-correlation seems a bit strict. What was the electrode spacing of the intracranial grids and stereo EEG? How many tested pairs remained after applying the rule?

Page 24 Line 6: The parentheses seem to be redundant in the equation.

We thank the reviewer for pointing out. The extra parentheses have been removed.

• Is the statistical analysis of the data sound?

Yes, if the classification of the data was done on unseen test data.

Page 13 Line 4/5: “To evaluate the contribution of different aspects of the endogenous features, the same model was applied using different subsets of the pre-stimulus features.” Repeated tests with mixed features require a correction, in the most conservative case a Bonferroni correction.

We thank the reviewer for pointing this out. We did not include an explicit correction in the first place because the p-values of the paired are much smaller than 0.001. Considering there were less than 10 repeated tests performed, we just reported uncorrected $p < 0.001$. The p-values are still much smaller than 0.001 after taking Bonferroni correction for 7 repeated tests, and we have now reported the Bonferroni corrected p-values in page 14 (line 10).

• Should the authors be asked to provide further data or methodological information to help others replicate their work? (Such data might include source code for modelling studies, detailed protocols or mathematical derivations).

Yes, the methods have to be explained and clarified.

• Are there any special ethical concerns arising from the use of animals or human subjects?

Data were recorded from patients who underwent epilepsy monitoring and thus, required intracranial electrodes for clinical reasons. The study protocol was approved and the patients were informed.

Reviewers' Comments:

Reviewer #1:

Remarks to the Author:

The authors have made a great effort in revising their manuscript. I appreciate especially the modifications in making the (still complicated) methods more accessible and comprehensible. I can recommend this manuscript for publication.

Reviewer #2:

Remarks to the Author:

The authors have adequately addressed many of my concerns. I will note that I do not have sufficient expertise with multivariate logistic regression models to judge whether the details outlined in the Supplementary Methods are adequately implemented. However the addition of the 'cartoon' (Fig 1C) and the pseudocode algorithm are helpful in making the methods more accessible and giving the general reader an intuition of the approach. NB, in consideration of improved transparency and reproducibility (and also in light of the potential applications of this modeling approach in other domains, highlighted in the revised Discussion) it would be even better if the authors would make the actual code openly available, e.g. on Github, rather than only on request.

Below I outline a number of mostly minor remaining concerns and request for clarifications.

Major comments:

The only main concern I still have is about MI-RT relationship (Results reported in Figures 2B-C). First a small point (also to adhere with the journals requests about reporting n for each analysis): as I noted before I think it should be clearer in the main text on how many trials these estimates are based: the captions mention the number of electrodes, which remain the same across for each analysis, but the point is that the number of trials these estimates are based on is much reduced relative to (for example) figures 2A and 2D, because the number of repeated trials is 120 rather than 480, and splitting that up further in 6 categories leaves only 20 trials to estimate the RT for 'preferred' category trials, I think(?), not taking account rejected trials in preprocessing etc.

Second, most or all of the other analyses report a comparison between modeling outcomes obtained using post-stimulus activity only versus including pre-stimulus activities/MIs, which provides evidence for the added value of including pre-stimulus activity in decoding. But for these RT analyses it's in fact not clear to me whether you would be able to find a trial-by-trial correlation between the neural activity and RT based on the post-stimulus activity only, and if so, if this correlation is 'significantly improved' if you include the prestimulus activity. In other words, does the MI uniquely predict additional behavioral variance, or are you maybe 'piggy-backing' on a relationship between category-selectivity and behavior, whereby electrodes with strong category-selective responses will be predictive of RT on trials of that category? For example, could it be the case that you systematically get a higher MI (i.e. a bigger 'shift' in the decision boundary) for trials with a weak category-selective response that have a long RT?

This concern is related to the second point I made in my previous review under 'Results' ("[..] I wonder if the lack of an RT effect for non-preferred stimuli may be related to the response to the non-preferred stimuli being low magnitude overall, or more variable across trials?"). To this question the authors responded that "we are not examining the correlation between the stimulus response and behavior, but rather the prestimulus MI and behavior". While I understand that the MI is feature of

interest here, my point is that it in fact might be worth looking into the other correlation (between stimulus response and behavior) because I'm not sure to what extent the relation between prestimulus MI and behavior can be explained by the relation between the response itself and behavior. (This is also relevant for supporting the claim on Page 17, line 22-24, that "the results demonstrated a significant relationship between the pre-stimulus MI and the reaction time in detecting repetitions in the category that the electrode is selective for"). Can the authors look into this, or perhaps alternatively, explain how this is already taken into account in the modeling?

Minor comments:

Methods:

As I mentioned above the pseudocode and algorithm help in making the approach more understandable. However I still got a bit stuck on the fact that Eq 1 is expressed as the 'argmax' form, i.e. as an expression about the minimization/loss function, followed by a lot of detail about the regularization parameters; for me, it would be helpful to start out with a formula that expresses a relationship between a Y (classification) and an X (input features) and the weights on those features (betas), which can then be followed by how those weights are then estimated (the loss function). I realize the authors may consider this to be textbook material or too simplistic, but I think it would be helpful for a reader who is familiar with 'simpler' regression models/GLMs to ease into this topic. Currently, the features that are being classified (X) are not introduced until a couple of paragraphs later at the bottom of page 26 ("We consider the neural activity within..."). I suggest moving those up to the start of the methods section, followed by an explanation of the logistic regression itself (even if textbook style) and then the loss function and associated details.

Beeswarm plots:

It's great to have the individual data points in the plots, but the size at which they are plotted now they were nearly invisible on print, particularly so in Figure 2 (I initially didn't even notice them!). Please adjust the size and/or transparency to make the data more visible. Also, for Figures 2C-D it would be helpful if the datapoints would be color-coded by category (e.g. using the same code as for the electrode recon figure in 1B) so we can see whether for example the MI-RT correlations are more pronounced for e.g. face versus other stimulus categories.

Discussion:

For a main conclusion statement, the last sentence of first paragraph (line 10-12) still was too technical for me (it's also not exactly the same as what is mentioned in rebuttal). In particular I find the phrase '[...] along the critical category selective dimensions' hard to follow, I'm not sure what the word 'critical' is supposed to express and how you've excluded a possible modulation that happening 'orthogonal' or 'oblique' to the decision boundary, for example.

Page 19, lines 2-4: having a concrete example of the contexts that are sketched here would be really helpful, for example, what type of stimuli would you be looking for and how would the endogenous state facilitate their processing?

Figures + small textual suggestions:

Page 5, line 6: insert 'grayscale' before 'images'.

Page 4, line 21: 'examined' > 'extracted', or 'computed'

Page 6, line 1: emphasized 'that'

Page 6, line 5/6: this sentence sounded bit weird to me, I think the issue is that to say that 'prestimulus activity contains information about the response' makes it sound as if the brain is

somehow prescient about the upcoming stimulus.

Figure 1B: why not also show the lateral view of the RH?

Figure 1B: the caption mentions a second criterion based on response magnitude for determining category-selectivity that is not described in the Methods section.

Figure 1C, right panel: what are the transparent and opaque dots supposed to represent?

Figure 1C: the caption mentions a dashed red line while the lines are either black or very dark gray. It would be helpful to distinguish them more clearly visually and to make sure to describe each element in the figure in the caption.

Page 10, line 17: 'the' is missing before the word 'mean' (2x)

Page 12, line 25: across brain > across brain regions

Page 15, line 24 + Figure 3C caption: what is the 'sparse' GLM, does this refer to a subset of the two-stage GLM?

In general, some of the new text (in blue) had some grammatical typos (e.g. missing words like 'the' etc), please double check

In the Supplementary Methods, some of the notation appears to be different from the main manuscript (e.g. the BHA is referred to as BB and stP as ERP).

Reviewer #3:

Remarks to the Author:

The authors have resolved all concerns raised in the review and the methods appear now to be appropriately implemented. Important is that results have been obtained after cross validation. Also the role of the reaction time in the study is now clear.

The authors acknowledged in the discussion that it is likely, but not guaranteed, that perception has been influenced. This is an appropriate inference, but now contradicts with the title that clearly states that endogenous activity modulates perception.

This should be removed from the title as it could be misleading.

REVIEWER COMMENTS

Reviewer #1 (Remarks to the Author):

The authors have made a great effort in revising their manuscript. I appreciate especially the modifications in making the (still complicated) methods more accessible and comprehensible. I can recommend this manuscript for publication.

We thank the reviewer for the supportive remarks and efforts with our work.

Reviewer #2 (Remarks to the Author):

The authors have adequately addressed many of my concerns. I will note that I do not have sufficient expertise with multivariate logistic regression models to judge whether the details outlined in the Supplementary Methods are adequately implemented. However the addition of the 'cartoon' (Fig 1C) and the pseudocode algorithm are helpful in making the methods more accessible and giving the general reader an intuition of the approach. NB, in consideration of improved transparency and reproducibility (and also in light of the potential applications of this modeling approach in other domains, highlighted in the revised Discussion) it would be even better if the authors would make the actual code openly available, e.g. on Github, rather than only on request.

We thank the reviewer for the positive remarks. We have now included a demo code of the algorithm with sample data on GitHub (<https://github.com/yuanningli/two-stage-GLM>).

Below I outline a number of mostly minor remaining concerns and request for clarifications.

Major comments:

The only main concern I still have is about MI-RT relationship (Results reported in Figures 2B-C). First a small point (also to adhere with the journals requests about reporting n for each analysis): as I noted before I think it should be clearer in the main text on how many trials these estimates are based: the captions mention the number of electrodes, which remain the same across for each analysis, but the point is that the number of trials these estimates are based on is much reduced relative to (for example) figures 2A and 2D, because the number of repeated trials is 120 rather than 480, and splitting that up further in 6 categories leaves only 20 trials to estimate the RT for 'preferred' category trials, I think(?), not taking account rejected trials in preprocessing etc.

The reviewer is correct that results in Fig 2B-C are based on a reduced set of trials, compared to Fig 2A and 2D. This is also why larger variance was seen in the preferred conditions in Fig 2B-C, compared to 2A and 2D. We have added the following note to the caption of Fig 2 in the Results: "for the preferred conditions in Fig. 2B and 2C, only 1/6 of the repeated trials were included in the analysis (1 out of 6 categories and 20% of trials were repeat, so on average 20 trials in each block per subject per electrode). As a result, larger variance for preferred conditions was seen in Fig. 2B and 2C, compared to the rest of Fig. 2." (Page 13, Lines 2-6)

Second, most or all of the other analyses report a comparison between modeling outcomes obtained using post-stimulus activity only versus including pre-stimulus activities/MIs, which provides evidence for the added value of including pre-stimulus activity in decoding. But for these RT analyses it's in fact not clear to me whether you would be able to find a trial-by-trial correlation between the neural activity and RT based on the post-stimulus activity only, and if so, if this correlation is 'significantly improved' if you include the prestimulus activity. In other words, does the MI uniquely predict additional behavioral variance, or are you maybe 'piggy-backing' on a relationship between category-selectivity and behavior, whereby electrodes with strong category-selective responses will be predictive of RT on trials of that category? For example, could it be the case that you systematically get a higher MI (i.e. a bigger 'shift' in the decision boundary) for trials with a weak category-selective response that have a long RT?

We thank the reviewer for bringing up this concern, which allows us to further clarify our results. To some extent, this cannot be fully disentangled because the MI is an estimate of how much the post-stim is modulated by the pre-stim. Thus, while the correlation between the MI and behavior measures the relationship between “the aspect of pre-stimulus activity that modulates neural decoding accuracy and perceptual behavior” this is very similar to asking if the aspect of the post-stimulus that is modulated by the pre-stimulus activity is correlated with behavior (though not exactly equivalent because the model is not fully invertible). The closest that can be done is examining if all of the post-stimulus activity is more correlated to behavior than the MI or not. To address this concern, we computed the mean correlation coefficient between the post-stimulus response loading in the classifier (i.e. $X_{post}\beta_{post}$) and the corresponding RT. We found that even for the trials in the preferred category of the electrode, there is no significant correlation between the post-stimulus loading and the RT, significantly lower than the correlation between the MI and the RT (mean Spearman' rho for post-stim = -0.019, comparison between MI-RT correlation and post-stim-RT correlation $t(245) = 2.76, p = 0.006$, paired two-sided). This suggests that the aspect of the post-stim discriminant activity modulated by the pre-stim is more correlated to behavior than the overall post-stim activity is. Thus, it is unlikely that our findings are due to piggy-backing on a general relationship between category-selectivity and behavior and instead the MI-behavior correlation is a distinct aspect of the neural activity related to behavior. We thank the reviewer for this comment as it has allowed us to add an interesting nuance to the interpretation of the results. (Page 10, Lines 11-20)

This concern is related to the second point I made in my previous review under 'Results' (“[...] I wonder if the lack of an RT effect for non-preferred stimuli may be related to the response to the non-preferred stimuli being low magnitude overall, or more variable across trials?”). To this question the authors responded that “we are not examining the correlation between the stimulus response and behavior, but rather the prestimulus MI and behavior”. While I understand that the MI is feature of interest here, my point is that it in fact might be worth looking into the other correlation (between stimulus response and behavior) because I'm not sure to what extent the relation between prestimulus MI and behavior can be explained by the relation between the response itself and behavior. (This is also relevant for supporting the claim on Page 17, line 22-24, that “the results demonstrated a significant relationship between the pre-stimulus MI and the reaction time in detecting repetitions in the category that the electrode is selective for”). Can the

authors look into this, or perhaps alternatively, explain how this is already taken into account in the modeling?

In general the reviewer is likely correct in that the correlation between MI and behavior for the non-preferred is low due to a relatively small post-stim response for the non-preferred categories. The key point about the low MI-behavioral correlation for the non-preferred condition is that it provides supportive evidence that the MI effect we report is not a global, non-specific effect like arousal or alertness. This can be thought of as an independent result that illustrates a similar point as the low correlation of the MI between electrodes that prefer different categories. For example, if the MI effect were global, one would expect that the MI would correlate between electrodes regardless of category preference and that the MI would correlate with behavior for all categories. The result that neither of these things are true support our interpretation that the endogenous effects we report are specific and not global.

In looking at the previous version of the manuscript, we realize that we perhaps did not clearly indicate what the lack of MI-behavioral correlation for the non-preferred condition implies about the pre-stimulus modulation. We have tried to make clear on Page 10 that the main thing this lack of correlation shows is that the effects we report are not global and non-specific.

Minor comments:

Methods:

As I mentioned above the pseudocode and algorithm help in making the approach more understandable. However I still got a bit stuck on the fact that Eq 1 is expressed as the 'argmax' form, i.e. as an expression about the minimization/loss function, followed by a lot of detail about the regularization parameters; for me, it would be helpful to start out with a formula that expresses a relationship between a Y (classification) and an X (input features) and the weights on those features (betas), which can then be followed by how those weights are then estimated (the loss function). I realize the authors may consider this to be textbook material or too simplistic, but I think it would be helpful for a reader who is familiar with 'simpler' regression models/GLMs to ease into this topic. Currently, the features that are being classified (X) are not introduced until a couple of paragraphs later at the bottom of page 26 ("We consider the neural activity within..."). I suggest moving those up to the start of the methods section, followed by an explanation of the logistic regression itself (even if textbook style) and then the loss function and associated details.

We thank the reviewer for this very helpful suggestion. We agree that the suggested adjustment would make the general approach clearer. We have made the suggested changes in the Methods section (Pages 25-26).

Beeswarm plots:

It's great to have the individual data points in the plots, but the size at which they are plotted now they were nearly invisible on print, particularly so in Figure 2 (I initially didn't even notice them!). Please adjust the size and/or transparency to make the data more visible. Also, for Figures 2C-D it would be helpful if the datapoints would be color-coded by category (e.g. using the same code as for the electrode recon figure in 1B) so we can see whether for example the MI-RT correlations are more pronounced for e.g. face versus other stimulus categories.

We thank the reviewer for the helpful suggestions. We have adjusted the size of the data points in Figure 2. We also color-coded the dots in Fig 2A-D by category, using the same code as Fig 1B.

Discussion:

For a main conclusion statement, the last sentence of first paragraph (line 10-12) still was too technical for me (it's also not exactly the same as what is mentioned in rebuttal). In particular I find the phrase '[...] along the critical category selective dimensions' hard to follow, I'm not sure what the word 'critical' is supposed to express and how you've excluded a possible modulation that happening 'orthogonal' or 'oblique' to the decision boundary, for example.

We thank the reviewer for raising this point. We agree that our results did not fully exclude the possible modulation that happening orthogonal or oblique to the decision boundary. We have changed this sentence into "The results here demonstrate that pre-stimulus activity modulates the post-stimulus activity in the regions that are selective for the stimulus being viewed." (Page 17, Lines 12-13)

Page 19, lines 2-4: having a concrete example of the contexts that are sketched here would be really helpful, for example, what type of stimuli would you be looking for and how would the endogenous state facilitate their processing?

We thank the reviewer for the suggestion. We have changed the sentence into "While in the present study a strategic process would not provide a behavioral advantage, for visual perception in familiar environments that one commonly finds oneself in, such as one's house or office, facilitating the processing of particular stimuli may be advantageous." (Page 19, Lines 3-6)

Figures + small textual suggestions:

Page 5, line 6: insert 'grayscale' before 'images'.

Page 4, line 21: 'examined' > 'extracted', or 'computed'

Page 6, line 1: emphasized 'that'

We thank the reviewer for these suggestions. We have made the corresponding changes.

Page 6, line 5/6: this sentence sounded bit weird to me, I think the issue is that to say that 'prestimulus activity contains information about the response' makes it sound as if the brain is somehow prescient about the upcoming stimulus.

We thank the reviewer for raising this concern. We have changed the sentence into "Because the pre-stimulus activity contains no information about the upcoming stimulus category (see supplemental results), classification accuracy can be improved using this model only if the pre-stimulus activity contains information about the conditional distribution of the post-stimulus response on a particular trial (e.g. larger/lower variance, gain, etc.)." (Page 6, lines 3-7)

Figure 1B: why not also show the lateral view of the RH?

We thank the reviewer for the suggestion. We have added the lateral view of the RH to Fig 1B.

Figure 1B: the caption mentions a second criterion based on response magnitude for determining category-selectivity that is not described in the Methods section.

We thank the reviewer for pointing this out. We have added descriptions of this constraint to the Methods section: "To avoid the rare case where an electrode showed visual response to all but one category, we add additional constraint that the assigned category should have larger event-related potential (mean stP) or broadband high-frequency activity (mean stBHA) over other categories." (Page 25, lines 13-16)

Figure 1C, right panel: what are the transparent and opaque dots supposed to represent?

Figure 1C: the caption mentions a dashed red line while the lines are either black or very dark gray. It would be helpful to distinguish them more clearly visually and to make sure to describe each element in the figure in the caption.

We thank the reviewer for the useful suggestions. We have added additional descriptions of the elements in Fig 1C in the caption.

Page 10, line 17: 'the' is missing before the word 'mean' (2x)

Added.

Page 12, line 25: across brain > across brain regions

Added.

Page 15, line 24 + Figure 3C caption: what is the 'sparse' GLM, does this refer to a subset of the two-stage GLM?

We thank the reviewer for pointing this out. The sparse GLM was actually referring to the two-stage GLM since the weights in the model are sparse. We have changed it into just 'two-stage GLM' to avoid confusion.

In general, some of the new text (in blue) had some grammatical typos (e.g. missing words like 'the' etc), please double check

We thank the reviewer for pointing this out. We have checked the new text and corrected the typos.

In the Supplementary Methods, some of the notation appears to be different from the main manuscript (e.g. the BHA is referred to as BB and stP as ERP).

We thank the reviewer for pointing this out. We have changed the notations in the Supplement.

Reviewer #3 (Remarks to the Author):

The authors have resolved all concerns raised in the review and the methods appear now to be appropriately implemented. Important is that results have been obtained after cross validation. Also the role of the reaction time in the study is now clear.

The authors acknowledged in the discussion that it is likely, but not guaranteed, that perception has been influenced. This is an appropriate inference, but now contradicts with the title that clearly states that endogenous activity modulates perception. This should be removed from the title as it could be misleading.

We thank the reviewer for the supportive remarks. We have now changed the title into "Endogenous activity modulates stimulus and circuit-specific neural tuning and predicts perceptual behavior"

Reviewers' Comments:

Reviewer #2:

Remarks to the Author:

The authors have addressed all of my concerns